# Cecelia: a multifunctional image analysis toolbox for decoding spatial cellular interactions and behaviour

Dominik Schienstock ⬤ [1], Jyh Liang Hor[1,2], Sapna Devi[1] & Scott N. Mueller ⬤ [1] ✉

With the ever-increasing complexity of microscopy modalities, it is imperative to have computational workflows that enable researchers to process and perform in-depth quantitative analysis of the resulting images. However, workflows that allow flexible, interactive and intuitive analysis from raw images to analysed data are lacking for many experimental use-cases. Notably, integrated software solutions for analysis of complex 3D and live cell images are sorely needed. To address this, we present Cecelia, a toolbox that integrates various open-source packages into a coherent data management suite to make quantitative multidimensional image analysis accessible for non-specialists. We describe the application of Cecelia to several immunologically relevant scenarios and the development of an unbiased approach to distinguish dynamic cell behaviours from live imaging data. Cecelia is available as a software package with a Shiny app interface (https://github.com/schienstockd/cecelia). We envision that this framework and its approaches will be of broad use for biological researchers.

Choosing appropriate workflows for image analysis is complex[1]. With the advent of clinically relevant multiplex imaging, many recent analysis pipelines focus on 2D multicolour images. While these techniques offer a unique insight into the spatial cellular composition for many immune responses, these are not the only types of images that are used to understand immune cell interactions in experimental settings. Studies have highlighted that the 3D organisation of tissues, including the lymphoid organs, is crucial for the efficient induction of immune responses[2–6]. For example, T cells must traverse complex 3D networks of fibroblasts to locate and interact with cognate antigen-presenting cells[7]. Although methods including tissue clearing have enabled robust 3D tissue imaging, analysis of the resulting imaging data remains challenging. One approach to analyse spatial tissue organisation in 3D is histo-cytometry[8–10], which involves significant manual processing and several unconnected software packages. There is no unified framework that supports spatial immune cell profiling from the raw data to the final figure in this setting.

Current approaches rely on a sequence of software tools that need to be manually connected, and/or custom analysis scripts that may be context-specific. These current approaches are labour-intensive, which can lead to reproducibility issues due to the high manual involvement and parameter decisions at each step. For example, a typical histo-cytometry pipeline for 3D images[9] involves segmentation with Imaris (Oxford Instruments) (0.5–4 h), often combined with ImageJ for cases with densely packed cells (2–24 h). These data must be exported as CSV files and imported into a flow cytometry package such as FlowJo (BD) for gating (2–4 h). For spatial quantification, the gated populations can be imported into CytoMAP to apply appropriate analysis methods[8]. Once the data is loaded into FlowJo, there is no connection back to the original image and the user cannot visualise spatial analysis directly on the image. Some of these shortcomings have been addressed by the commercial software StrataQuest (TissueGnostics), however this only supports 2D images.

While static imaging provides a comprehensive insight into the spatial cellular architecture of tissues, it does not yield knowledge about dynamic interactions between cells. Spatial juxtaposition does not necessarily relate to effective cell interactions. Live cell imaging, especially intravital imaging, closed this gap by enabling a glimpse into

[1]Department of Microbiology and Immunology, The University of Melbourne, The Peter Doherty Institute for Infection and Immunity, Melbourne, VIC, Australia.
[2]Present address: Lymphocyte Biology Section, Laboratory of Immune System Biology, NIAID, NIH, Bethesda, MD, USA. ✉e-mail: smue@unimelb.edu.au

temporal cell interactions within living animals. Despite the success of intravital imaging to disentangle immune cell dynamics during infection and other localised and systemic perturbations, there is a lack of analysis pipelines to quantify these comprehensively. Most studies investigate cells by quantifying cell speed and other tracking parameters, often averaged across populations of cells. However, there is an inherent heterogeneity in cellular behaviours which is not captured by these standard methods[11]. Semi-supervised methods to identify cell behaviours have been proposed using manual gating[12] or track clustering[13]. For instance, profiling of immune cell behaviours through the analysis of dynamic and morphological parameters was demonstrated using intravital imaging of myeloid cells[13]. Yet, such approaches do not currently provide straightforward software frameworks to apply the same principles to a broad range of experimental data.

In this work we demonstrate the utility of our developed open-source software toolkit, Cell-cell interaction analyser (Cecelia), which enables quantitative analysis of multidimensional fluorescent microscopy images within a user-friendly interface. By employing Cecelia to diverse multiplexed static and live immunofluorescence imaging data, we demonstrate the broad practical utility of our solution for analysis of cellular interactions and behaviours in healthy tissues and during disease.

## Results

### Cecelia as a general-purpose image analysis framework
We developed Cecelia to integrate static and live cell image analysis with modular workflows in a unified framework with a graphical user interface (GUI). This toolbox combines R/Shiny and the image viewer napari[14], enabling interactive visualisation of data analyses on the microscopy images (Fig. 1a–c and Supplementary Fig. 1a, b; more detailed descriptions in 'Methods'). Cecelia is a centralised management system that allows reproducible and customised workflow design from raw images to data quantification (Fig. 1a, b). Cecelia is available as an R package (https://github.com/schienstockd/cecelia) and Docker container (https://github.com/schienstockd/ceceliaDocker). Documentation and tutorials are available at https://cecelia.readthedocs.io/. A step-by-step walkthrough of the analysis for each image in this article is at https://cecelia.readthedocs.io/en/latest/missions.html.

The proposed framework accepts all Bio-Formats (https://www.openmicroscopy.org/bio-formats/) supported images, which are imported into Cecelia and converted to the recently developed Open Microscopy Environment Next Generation File Format (OME-NGFF)[15]. Users can add metadata and clean up images before Cecelia guides the user through image segmentation, population definition, spatial analysis, and data plotting (Fig. 1d–f and see 'Methods'). Furthermore, to expand Cecelia's functionality, it is possible for users with basic programming experience to integrate processing and analysis workflows in the framework by creating custom modules (https://cecelia.readthedocs.io/en/latest/create_custom_module.html) or to work directly within R or Python notebooks.

### Spatial analysis of cellular interactions in complex 3D images using Cecelia
To evaluate the broad utility and flexibility of Cecelia, we applied the software to a range of imaging datasets (from multidimensional high-plex static to dynamic intravital two-photon imaging). The computational packages underlying these workflows are specialised Python and R packages to demonstrate the capability of Cecelia to integrate packages for custom analysis modules. A key strength of the Cecelia workflow is that it allows independent processing of channels and cell types for image segmentation which can be merged into a single segmentation to process images with mixed cell sizes, shapes, markers and reporters. This combined analysis of images containing a complex mix of endogenous fluorescence and antibody labelling with diverse cell morphologies is currently, to our knowledge, not possible with any single image processing framework.

We applied this combined segmentation workflow to 3D spleen tissue images from XCR1-venus mice that contain fluorescent conventional type I dendritic cells (DCs) and fluorescently labelled lymphocytic choriomeningitis virus (LCMV)-specific TCR transgenic P14 CD8$^+$ T cells, as well as immunofluorescence labelling with antibodies against T and B cells and virus infected cells (Fig. 2a). These images therefore contained diverse cell shapes and sizes as well as both membrane and cytoplasmic fluorescence. Segmentation of individual channels was performed with Cecelia using a Cellpose cytoplasm model (cyto2)[16] for lymphocytes that have rounded morphologies and a custom-trained Cellpose model for irregularly shaped DCs and LCMV-infected cells. These segmented channels were then merged into a combined segmentation for downstream analysis (Fig. 2b). Cell populations were manually gated by histo-cytometry (Fig. 2c) and individual cell and virus interactions quantitated by Delaunay triangulation neighbour detection (Fig. 2d). Clustering P14 T cells were identified using DBSCAN[17], a density-based clustering algorithm that identifies groups of coherent objects within a certain diameter (20 μm) and minimum number of elements (4 cells). An interaction between two cells was defined as the closest neighbouring cell from the Delaunay triangulation. Our spatial analysis revealed that clustering P14 T cells preferentially interacted with XCR1$^+$ DCs early during virus infection in the spleen, whereas interactions with LCMV-infected cells involved non-clustering T cells. These analyses demonstrate the capacity of Cecelia to perform quantitative spatial analysis of 3D images containing complex cell shapes and delineate cellular interactions.

### Quantitative analysis of highly multiplexed 3D images using Cecelia
To highlight the utility of combined structural and cellular segmentation of 3D images, we analysed a publicly available IBEX (iterative bleaching extends multiplexity) dataset from healthy human spleen[18] (Fig. 3a and Supplementary Fig. 1c). IBEX is a method to increase the number of fluorophores that can be imaged on samples by performing iterative staining and bleaching[18]. This requires a common marker across imaging cycles for subsequent image registrations, for example a nuclear marker such as Hoechst or DAPI. Current protocols use custom scripts and the commercial Imaris software for analysis and SimpleITK (https://simpleitk.org/) for channel alignment[18]. To analyse the IBEX-generated images, 3D image stacks were registered based on nuclei staining using Cecelia, and cells were segmented based on membrane markers and Hoechst resulting in a combined membrane and nuclear segmentation. The segmented objects were subjected to Leiden clustering for unbiased cell population identification (Fig. 3a). Leiden clustering is a community detection algorithm[19] that attempts to maximise the difference between groups while maintaining local cohesion between elements. Cell regions were extracted by first generating cellular neighbourhoods using a radius of 50 μm around each cell. K-Means clustering[20] was used to cluster these neighbourhoods into 3 regions. K-means clustering is a method to partition data points based on a predefined number of centre points. The aim is to categorise points by minimising the variance within in each partition. This method resulted in B cell, T cell and macrophage-rich regions. Structural markers, including Vimentin and CD163, were used to segment the stromal network with a custom Cellpose model. To quantify where cells were residing within the stromal network, the nearest stroma branch to any one cell was determined using the k-nearest-neighbours (knn) package ($k = 1$) in R. The three cell regions correlated with the branching density of the stroma network with B cell regions having the least and macrophage regions the most stroma contact (Fig. 3a).

To further explore functional regions, we examined CODEX (co-detection by indexing)[21,22] images from healthy human patients (https://portal.hubmapconsortium.org/) (Fig. 3b). CODEX utilises DNA-conjugated antibodies and iterative imaging with chemical

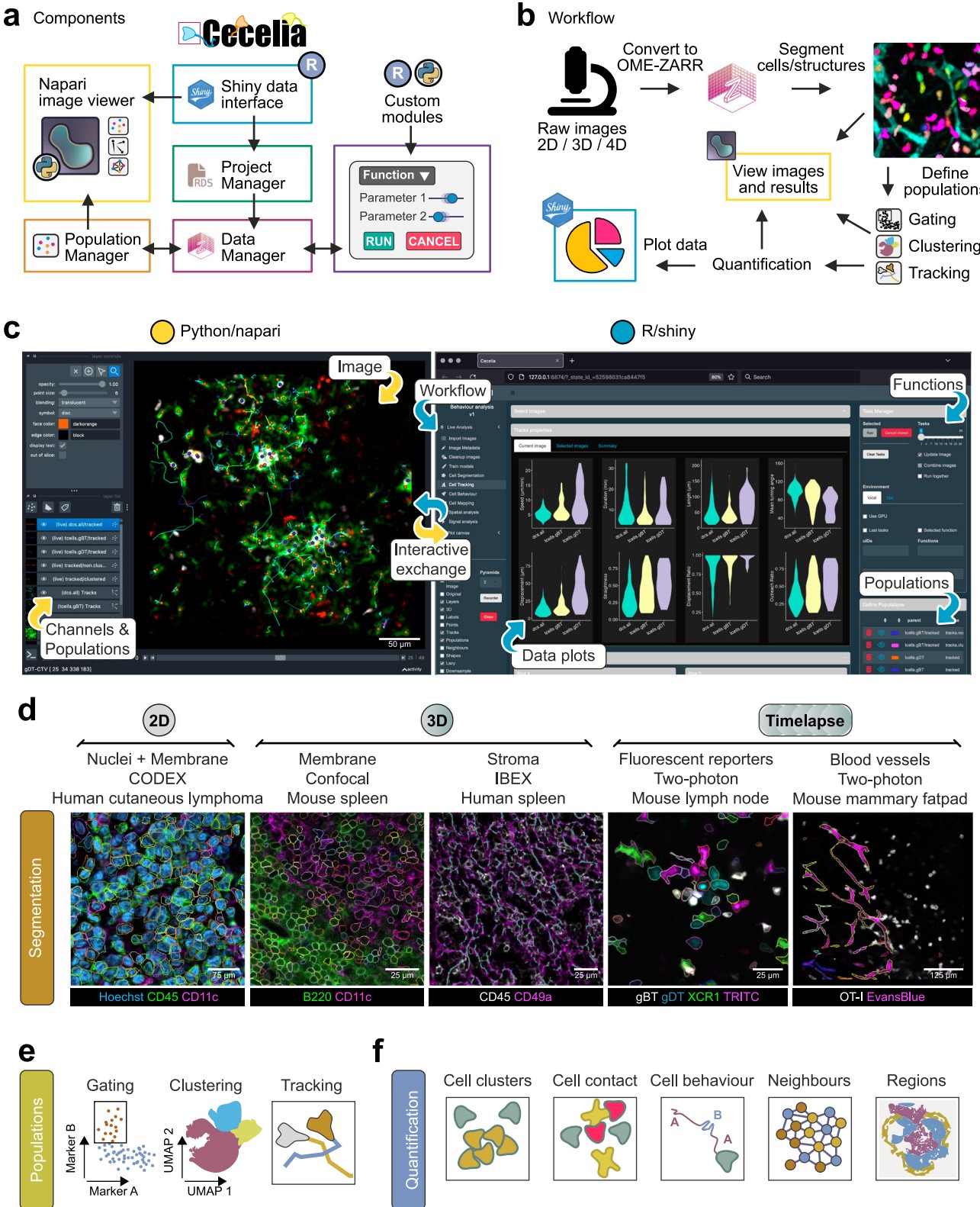

**Fig. 1 | Cecelia as a general-purpose image analysis framework. a** Simplified components diagram of Cecelia's main framework. See Supplementary Fig. 1b for a detailed version. **b** Overview of the Cecelia workflow. Images are converted to OME-Zarr, segmented and populations defined via various means. These populations can be spatially quantified, and the results visualised on the image itself and as data plots. **c** Overview of the graphical user interface. Python/napari and R/Shiny are the connecting partners that form the analysis framework. **d** Example images and segmentation results for various 2D, 3D and 4D imaging modalities to demonstrate the broad capability of the Cecelia framework. **e** Available analysis methods available to define populations in static and live cell images. **f** Main spatial analysis methods available for image quantification. This figure has been designed using resources from Flaticon.com and Fontawesome.com.

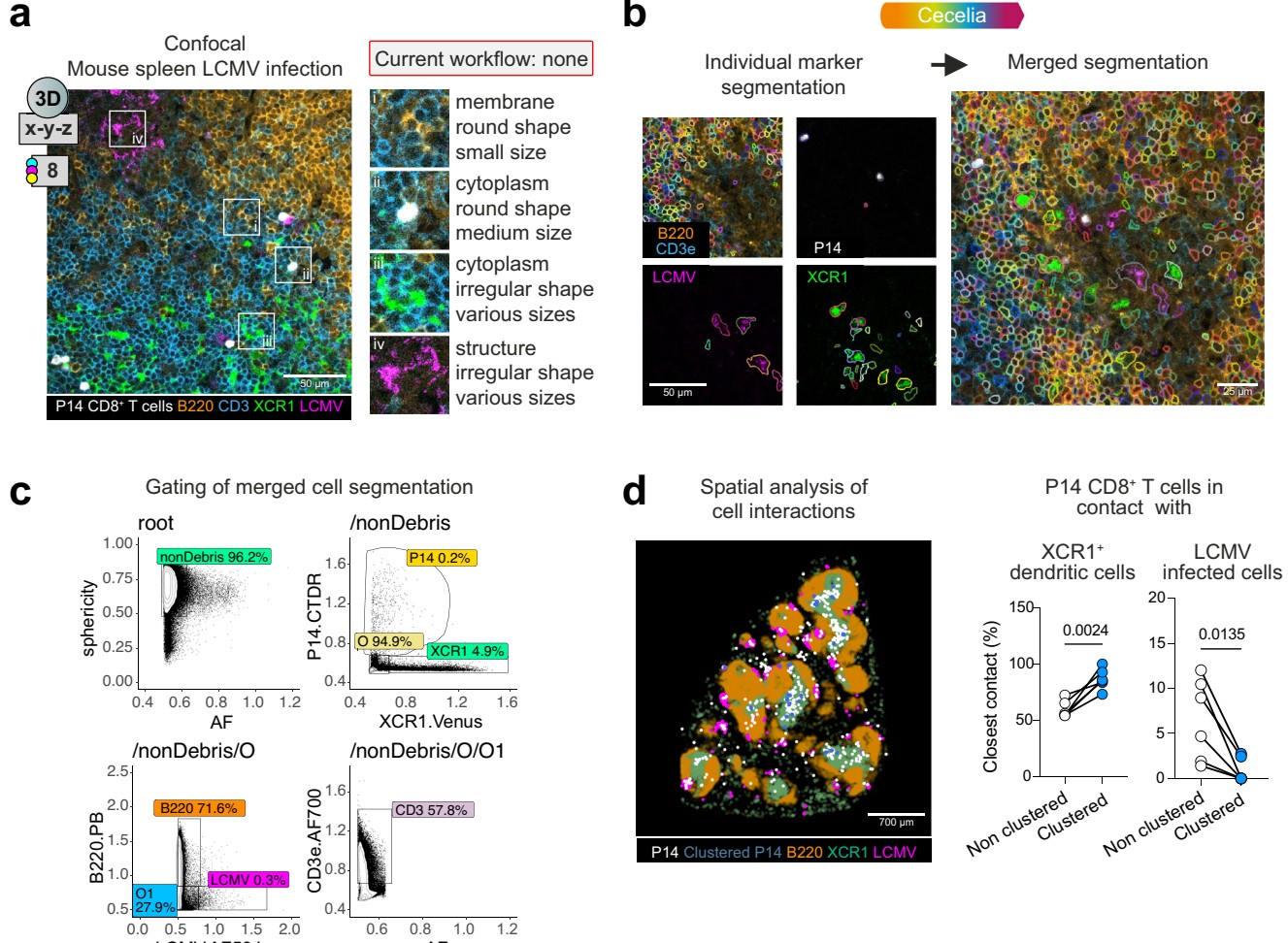

**Fig. 2 | Image segmentation and histo-cytometric quantification of 3D static images using Cecelia. a** Example confocal image containing diverse cell shapes and sizes and labels that cannot be segmented using current workflows. This 3D image contains 8 fluorescent channels (designated by the coloured circles). Cecelia's principle is to combine a mix of segmentations for combined spatial analysis. **b** Confocal imaging of mouse spleen 1.5 days after LCMV infection. Individual and merged segmentation of the individual markers is shown. **c** Histo-cytometry plots of merged image segmentation to identify cell types. **d** Spatial analysis and quantification of cellular interactions between virus specific CD8⁺ T cells, XCR1⁺ DCs and virus infected cells in mouse spleen 1.5 days after LCMV infection (*n* = 6, 2 independent experiments with 3 mice, two-sided *t*-test). Source data are provided as a Source Data file.

stripping to enable imaging of large marker panels[21]. The resulting images are often analysed using commercial software (i.e. HALO, Indica labs). Cecelia enables batch analysis of images for combined spatial cellular phenotyping. To demonstrate the applicability of Cecelia for analysis of multiplexed 2D images, we segmented 14 CODEX images of human lymph nodes and used Leiden clustering with subsequent Delaunay triangulation[23] to extract cellular neighbourhoods. These neighbourhoods were subjected to K-Means clustering to extract distinct regions enriched in T and B cells as well as lymphatics and blood vessels (Fig. 3b). This revealed the microanatomical patterns of cellular organisation that define human lymph nodes.

To demonstrate Cecelia's applicability to spatial data from diverse technologies, interactions between regions were also examined in a human breast cancer sample from Xenium in situ analysis[24] (Fig. 3c). Xenium is a fluorescent imaging technology that cycles through iterative DNA probe ligation to RNA targets, imaging and probe removal[24]. This technology allows researchers to identify RNA transcripts in fixed tissue samples. The resulting data are point clouds of transcripts with nuclear stains such as Hoechst. This transcript data was imported as individual image channels with a pixel summation to expand the transcript signal. We identified individual cell types, as above, using Leiden clustering. Cellular regions were

extracted using subsequent neighbourhood detection around each cell with a radius of 100 μm and K-Means clustering to extract 7 regions. Interactions between regions were extracted by mapping the closest cells to each other. Spatial analysis of tissue regions revealed tumour niche interactions, including regions enriched for CD4⁺ T cells and macrophages that were adjacent to invasive tumours but distal to regions of ductal carcinoma in situ (DCIS). These examples demonstrate the capacity of the Cecelia framework to be useful for detailed quantitative analysis of a wide variety of highly multiplexed 3D and 2D imaging modalities.

**Behavioural analysis of immune cell dynamics using Cecelia reveals hidden cell states**

The second component of our framework is the ability to comprehensively analyse live cell behaviour in a semi-supervised workflow. Dynamic cell behaviours and interactions are not captured during static imaging but can be captured using live imaging, including intravital imaging of cells in live animal models. Basic tracking measurements are commonly used for analysis of immune cells, including cell velocity, direction and angle. We illustrate an alternative workflow using Cecelia on intravital two-photon imaging data from the draining lymph nodes of mice during the response to cutaneous herpes simplex

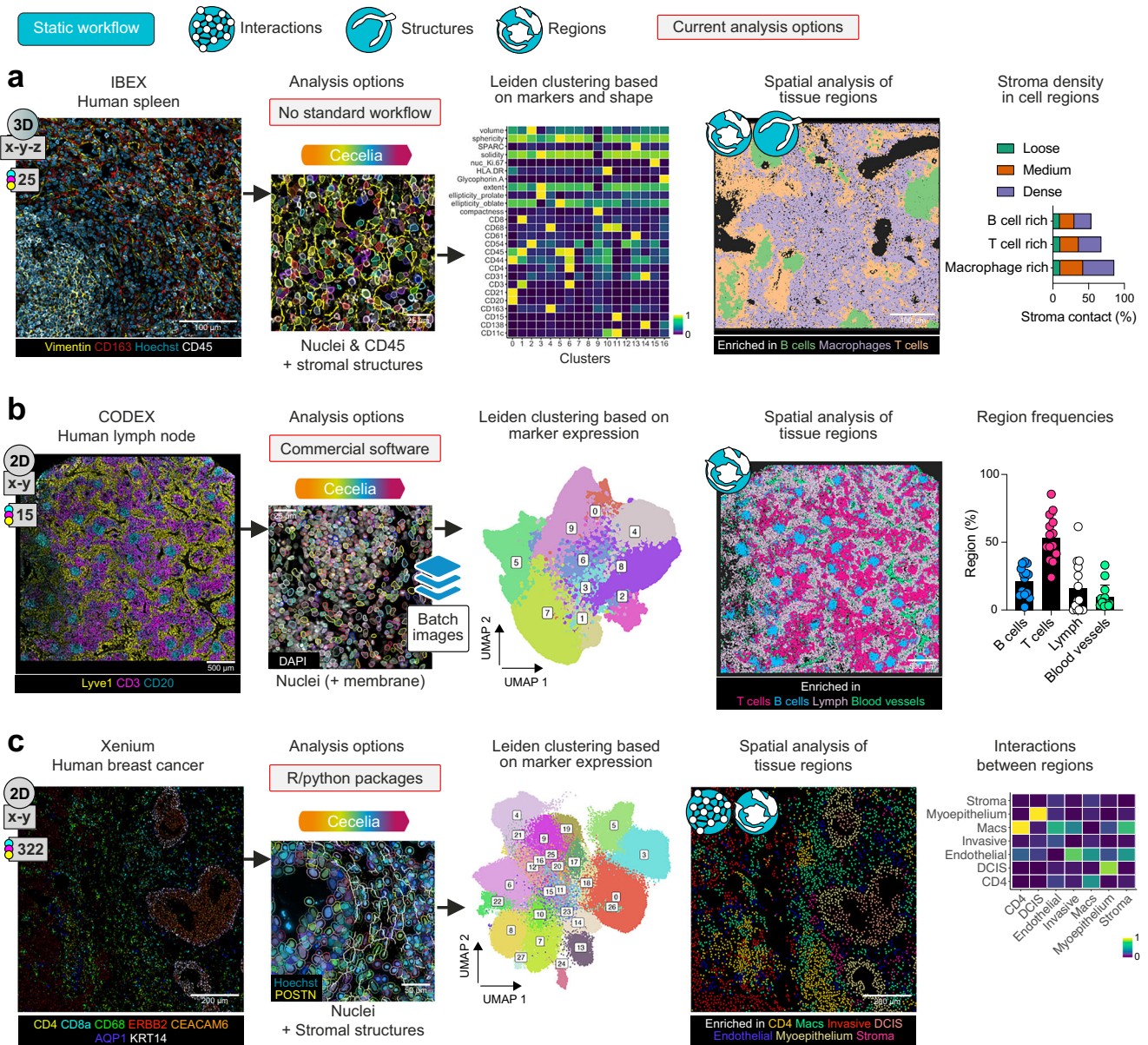

**Fig. 3 | Cecelia analysis of highly multiplexed 3D and 2D images.** Application of the Cecelia static image analysis workflow. **a** Analysis of images of healthy human spleen sections stained using the IBEX method (25 fluorescent channels). Only custom scripts are available currently to analyse IBEX images. Cecelia was used for image alignment and separate cell and stromal segmentation and Leiden clustering based on markers and cell shapes. Identified clusters are represented along x-axis of the heatmap. Analysis of interactions of immune cell regions with stromal networks and association with stromal density is shown. **b** Analysis of healthy human lymph nodes stained using CODEX ($n = 14$ images, 15 fluorescent channels, mean values ± SD). Commercial software is typically used for such analysis. Images were batch-processed in Cecelia cells clustered based on cell markers. Cell-type niche frequencies were analysed by spatial analysis of tissue regions. **c** Xenium in situ transcriptomic data obtained from human breast cancer samples was analysed using Cecelia (322 RNA probes as image channels). Cell region interactions were calculated from annotated cell type clusters. Values on the x-axis denote the regions from one to another region on y-axis. This figure has been designed using resources from Flaticon.com and Fontawesome.com. Source data are provided as a Source Data file.

virus (HSV) infection[3]. Upon infection of the skin or mucosal epithelium, HSV undergoes local replication and is controlled by virus-specific CD8⁺ and CD4⁺ T cell responses[25]. We found HSV-specific TCR transgenic gDT-II CD4⁺ T cells aggregating one day after infection by intravital imaging, indicating antigen-specific interactions with antigen-presenting cells that lead to T cell activation (Fig. 4a). Transgenic gBT-I CD8⁺ T cells aggregated one day later, on day 2 after infection, indicating temporal differences in T cell priming in lymph nodes during virus infection[3]. We could analyse these data in a conventional manner using common tracking statistics such as velocity that is averaged over time in each tracked cell (Fig. 4b). While the behavioural difference between the timepoints is captured by this

readout, demonstrated by a reduction in CD4⁺ T cell speed on day 1 and a subsequent reduction in CD8⁺ T cell velocities one day later, there is a significant heterogeneity in cell speeds observed even within naïve T cells which is not captured in this approach (Fig. 4b).

To better capture and interrogate the behavioural heterogeneity present amongst T cells migrating within tissues in live animals we developed a semi-supervised method to detect behaviour clusters with common tracking statistics but also behaviour states derived from Hidden Markov Models (HMM)[26,27] and their respective within-track transitions (Fig. 4c, d). To this end, we segmented T cells with Cellpose and tracked the segmented objects with the Python library btrack[28]. The resulting tracks were used for

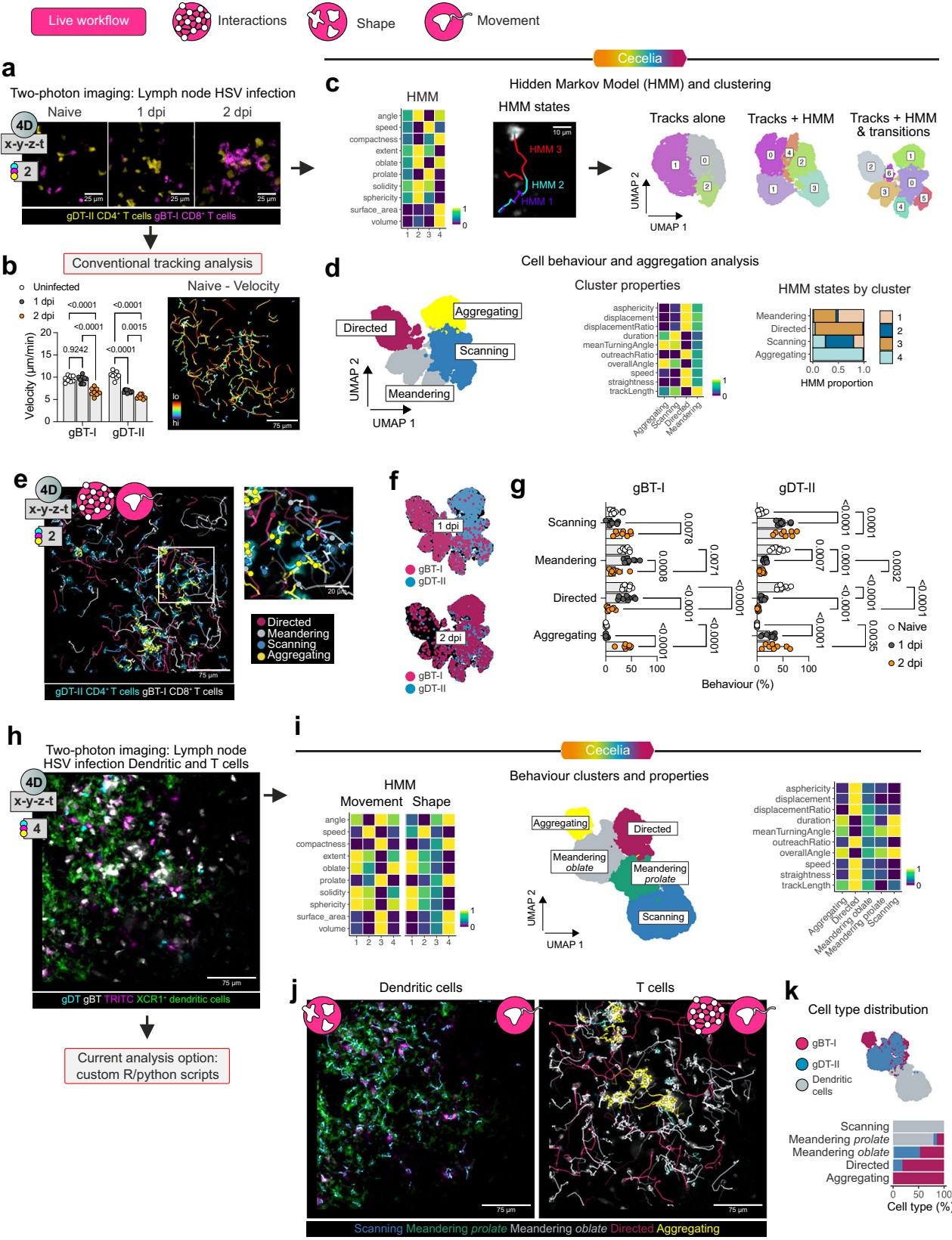

HMM analysis. HMM are statistical models that can be used to describe sequences of observations based on underlying variables that are inherent but may not be apparent in the data. We utilised speed and angle of the individual objects to identify HMM states (Fig. 4c). These states and their transitions were used in combination with track statistics for Leiden clustering. This method extracted considerably more refined behaviour patterns than tracking statistics alone (Fig. 4c).

As this dataset contained a significant number of aggregating cells, we utilised the Python library trimesh (https://trimesh.org/) to extract contacting cells (with a maximum distance of 5 μm). This analysis identified four distinct cell behavioural clusters amongst the

**Fig. 4 | Cecelia facilitates behavioural analysis of T cells and DCs in lymph nodes during virus infection. a–f** Analysis of T cell behaviour in lymph nodes draining the site of skin HSV infection. Representative frames of gDT-II CD4⁺ and gBT-I CD8⁺ T cells at different timepoints of infection: naïve mice and 1 d and 2 d after infection. **b** Velocity of tracked cells (*n* = 8–10 movies from 2–3 mice from at least 2 independent experiments, mean values ± SD, two-way ANOVA with Tukey's comparison test) (left, graph) and representative cell tracks coloured by velocity to demonstrate heterogeneity in cell movement. **c** Cell behaviour clustering was obtained by combining cell tracking data, hidden Markov models (HMM) and state transitions together for Leiden clustering. **d** Cell behaviour and aggregation were analysed. UMAPs of the resulting behaviour clusters and cell identities 1 dpi. **e** Representative image of T cells in the lymph node after infection and a magnified region. Cell tracks are coloured by cell behaviours (Supplementary Movie 1). **f, g** Plots depicting gBT-I and gDT-II cell behaviour distributions in uninfected mice and HSV infected mice (*n* = 8–10 movies from 2–3 mice from at least 2 independent experiments, mean values ± SD, two-way ANOVA with Tukey's comparison test). **h–k** Behavioural profiling of T cells and DCs in lymph nodes 2 days after HSV infection. **h** Imaging of gBT-I and gDT-II T cells, XCR1-venus DCs and TRITC-labelled antigen presenting cells. **i** HMM analysis of cell movement and shape (left panel), UMAP presentation of the behaviour clusters (middle panel) and heatmap of key tracking parameters defining the behavioural clusters (right panel). **j** Projection of cell tracks coloured by cell behaviour onto the images showing DCs (left) and T cells (right). **k** Distribution of cell types within the identified behavioural clusters. (*n* = 16 movies from 5 mice from 2 independent experiments). Source data are provided as a Source Data file.

responding T cells that we annotated as directed motion (high speed and track straightness and aspherical shape), meandering (intermediate speed, angle and cell shape), scanning (low speed and high turning angle and low asphericity) and aggregating in clusters (Fig. 4d, e). These behavioural clusters consisted of cells whose motion was dominated by one or two HMM states. These patterns can be extracted across multiple images and then compared. In this example, CD4⁺ and CD8⁺ T cell aggregation increased on days 1 and 2 post-infection, respectively, with the concomitant increase in scanning behaviour and decrease in meandering and directed motion (Fig. 4f, g).

We further investigated the utility of this approach by using Cecelia's behavioural analysis workflow to segregate different immune cell types within HSV-infected samples, DCs and T cells, by behavioural characteristics and shape differences alone (Fig. 4h–k). Published workflows[13] to separate immune cells by behaviour are currently a mixture of various independent scripts that the user must manually adjust for their needs. These also consider only whole track measurements which results, as shown above, in crude behaviour mapping. With HMM, Cecelia can combine movement, shape and individual cell interactions to comprehensively segregate cell types and their behaviour.

We first individually segmented gBT-I CD8⁺ T cells, gDT-II CD4⁺ T cells, XCR1⁺ DCs and migrating TRITC⁺ cells using Cellpose. The resulting objects were tracked using btrack (Supplementary Fig. 2). HMM cannot merely be utilised for movement patterns but for any measurement over time. In addition to the movement HMM, as utilised above, we used a combination of morphological parameters to extract 4 shape states (Fig. 4i). We utilised the transitions between these states in combination with whole track measurements to deduce track clusters of movement and shape (Fig. 4i). These track clusters were able to distinguish DC and T cell behaviour (Fig. 4j) where DCs primarily have reduced motility (scanning) or a meandering prolate behaviour. T cells on the other hand exhibit, relative to DCs, more meandering oblate and directed behaviour in addition to their specific aggregation behaviour due to the virus infection (Fig. 4k).

### Integration of structural components and cellular flow into live cell quantification using Cecelia

Structures, such as lymphatic vessels, can influence how and where cells migrate in tissues. Akin to static imaging, we can incorporate cell types and structures of varying shapes and sizes with individual segmentations into one analysis across images. We analysed movies from a publicly available dataset utilising intravital two-photon imaging of the mouse ear during Delayed-Type Hypersensitivity (DTH) response by CD4⁺ T cells to ovalbumin[29] (https://www.immunemap.org/) (Fig. 5a). Incorporating the distance of T cells to lymphatics into the behavioural cluster analysis, as opposed to choosing an arbitrary distance cut-off (Supplementary Fig. 3a), revealed distinct modes of behaviour close to or away from these structures. Cells that migrated at a greater distance from lymphatics were found to display more directed motion, whereas cells closer to lymphatics underwent a scanning mode of motility as cells followed the lymphatic vessels (Fig. 5a).

Individual track measurements are informative yet cannot capture collective cellular motion. To investigate cells undergoing collective cellular motion with Cecelia, we utilised publicly available in vitro migration assays on collagen-coated imaging chambers from a breast cancer cell line, MCF10DCIS, that either expressed a control shRNA or a shRNA to target and silence the filopodial protein MYO10[30] (Fig. 5b). These assays are used to investigate migration speed and dissemination properties of cancer cells in different settings. After segmentation and tracking, there was a clear difference in straightness between the treatments with MYO10-shRNA cells exhibiting more directed motion compared to controls. We wanted to further verify this phenotype by extracting the collective motion of cells. To this end, we extracted a flow field, directionality, of cell tracks using ILEE[31]. These flows were assessed for anisotropy, non-uniform distribution, which was increased in MYO10 targeted samples indicating increased collective directed motion corroborating standard tracking measurements (Fig. 5b and Supplementary Fig. 3b). This combination of track clustering, structures and collective behaviour highlights diverse migration patterns in various imaging modalities that can be delineated using the Cecelia toolbox.

## Discussion

Fluorescent imaging techniques continue to evolve and drive fundamental and clinically relevant discoveries in biology. Quantitative analysis of images is paramount to unravel spatial information. The development of open-source software platforms that enable comprehensive analysis of multidimensional (3D and 4D), multiplexed imaging by biological researchers is vital. In this work, we demonstrate the utility of the powerful processing workflows for static and live cell imaging that we combined into the Cecelia toolbox. We developed Cecelia to facilitate advanced quantitative image analysis, incorporated a GUI and integrated the image viewer napari to provide a user-friendly platform for a wide range of users. Cecelia consists of a Shiny user interface and napari viewer (Supplementary Fig. 1a). Users interact with the modules in the user interface, while we also provide the capacity to employ custom modules which are implemented in R and/or Python depending on the functionality required. We employed the recently developed OME-NGFF file format for processing and storage of large multi-dimensional images. For large images that require substantial processing, Cecelia can be run on High-Performance Computing (HPC) systems providing greater computation, memory and graphics processing units (GPU).

We employed Cecelia to analyse confocal microscopy images containing complex mixtures of fluorescently labelled cells of varied sizes and shapes, for which we could find no established analysis workflow. Further, we found that Cecelia provides an integrated workflow for the analysis of multiplexed imaging methods by histo-cytometry and unbiased analysis of spatial tissue niches and cellular interactions.

To advance the capacity to interrogate the dynamic behaviours of immune cells migrating within tissues, we developed an approach by combining HMM with combinations of cell movement and shape parameters measured from analysis of intravital imaging data to

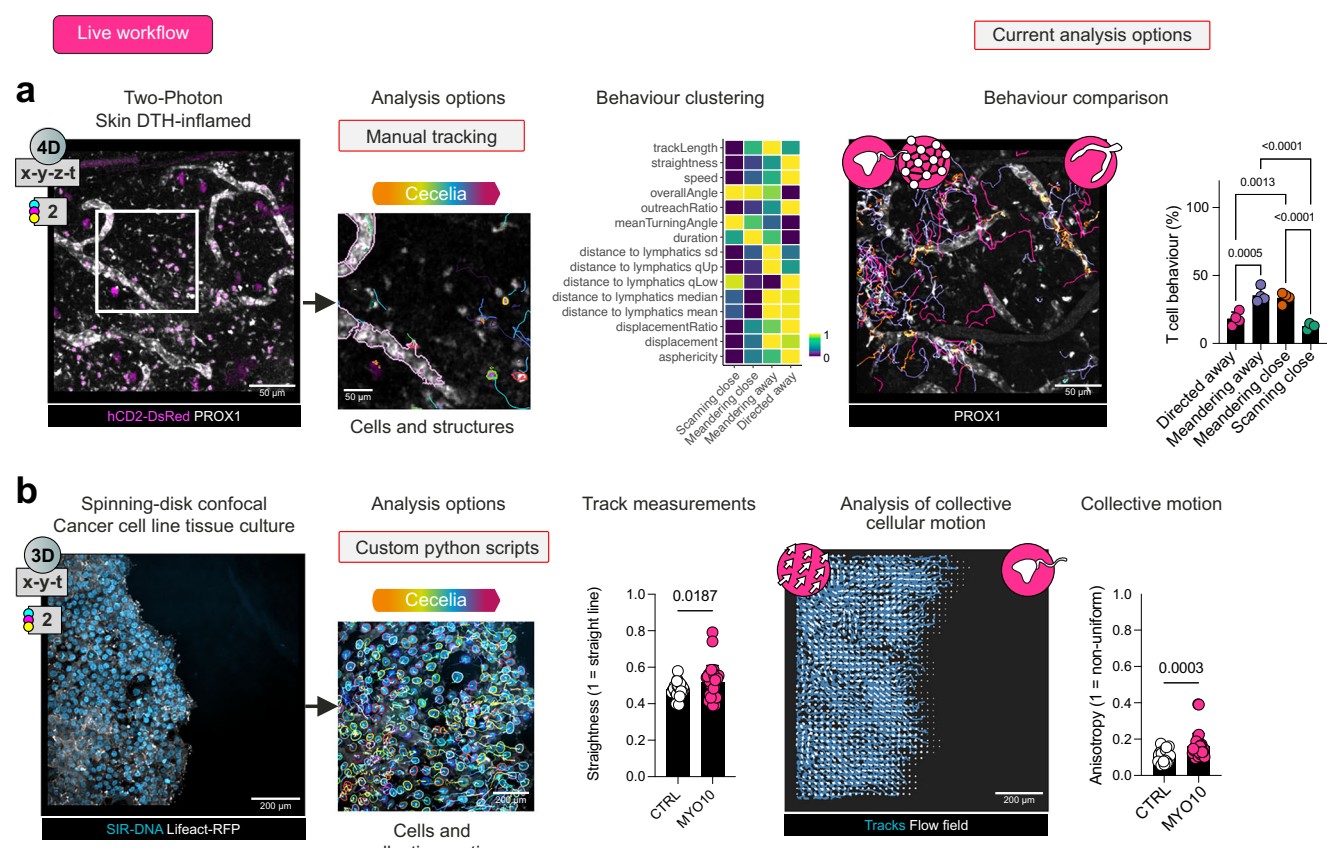

**Fig. 5 | Integration of structural components and cellular flow into live cell quantification. a** Cecelia analysis of the behaviour of T cells in DTH-inflamed skin identifies distinct behaviours by lymphatic vessel-interacting T cells. Independent segmentation of hCD2-DsRed expressing T cell cells and PROX1+ lymphatic vessel structures were generated and behavioural clustering performed on the T cells. Abbreviations for distance parameters shown in the heatmap: sd = standard deviation; qUp = 95% upper quantile; qLow = 5% lower quantile. Representative image of identified cell behaviours relative to the lymphatics, and quantitation of the proportion of behaviours ($n = 4$ movies, mean values ± SD, two-way ANOVA with Tukey's comparison test). **b** Flow field analysis of cancer cell migration imaged in vitro. MCF10DCIS lifeact-RFP tumour cells labelled with SiR-Hoechst were treated with CTRL shRNA or MYO10 shRNA and imaged by spinning disk confocal microscopy[30]. Straightness (total displacement divided by track length) (further tracking measurements shown in Supplementary Fig. 3b) of resulting tracks and Cecelia analysis to identify flow fields of collective cellular motion. Comparison of anisotropy (degree of inhomogeneity in collective directional motion) and cell behaviours by tumour cells treated with control or MYO10 targeted shRNA ($n = 29$–30 movies per group from 3 independent experiments, two-sided $t$-test, mean values ± SD). Source data are provided as a Source Data file.

classify cell behaviour. HMM takes measured parameters and fits a model to identify 'hidden behaviour' or 'hidden states' that are the underlying cause for the measured observations. HMM are therefore statistical models that can be used to describe sequences of observations based on underlying variables that are inherent but may not be apparent in the data. By incorporating this approach into Cecelia, we discovered that T cells migrating in lymph nodes in mice employ distinct behaviours that are altered during an immune response induced by virus infection. Furthermore, we show that Cecelia allowed identification of cell types based on unbiased assessment of cell movement and shape, distinguishing T cells from DCs. Although the identification of cells by behaviour is not absolute, as previously shown using a similar approach with custom analysis[13], this approach could be very useful when imaging of cells using distinguishing markers or dyes is limiting, such as intravital imaging in humans[32]. Nonetheless, our method is not limited to the specific behaviour types identified here, and annotation of cellular behaviours will likely depend upon the cell types and experimental conditions imaged. Importantly, we anticipate that the Cecelia toolbox and its proof-of-principle workflows will facilitate use of more comprehensive image analysis approaches by a broad range of biological researchers.

The use of R/Shiny, albeit a straightforward choice for our use cases (outlined in detail in 'Methods' section), is not the most performant framework for web applications and its combination with a Python backend forced us to create a solution that is more complex to install. Given this complexity and the usage of Anndata as a primary storage system, there can also be race conditions if Shiny and Python want to access the same dataset at the same time. The Shiny modules must therefore be thoughtfully designed when data is used in both environments. While it might be possible to implement the whole Cecelia framework in Python and therefore reduce some of these complications, it might not provide the same functionality as we would need to match processing that is currently being done in R with Python functions. Future developments could develop a solution that is a standalone executable based on the tools and processes that we are presenting in this article. Given the rise of artificial intelligence in many areas of image processing and analysis, as tools are developed or improved, our framework can be updated or new modules added. Our hope is therefore to provide a framework to create new interfaces and processing modules for this to be feasible.

## Methods
### Animals and ethics statement
Animal experiments followed the NHMRC Code of Practice for the Care and Use of Animals for Scientific Purposes guidelines and were approved by the University of Melbourne Animal Ethics Committee.

C57BL/6, gBT-I[33], gBT-I.uGFP, OT-I[34], OT-I.uGFP, P14[35], P14.ubTomato and XCR1-venus[36] mice were bred and housed in specific pathogen-free conditions in the Biological Research Facility (BRF) at the Doherty Institute for Infection and Immunity, the University of Melbourne. All animals were kept in HEPA filtered, individually ventilated cages, 19–22 °C, 40–70% humidity, with environmental enrichment, 12 h light-dark cycle and food and water *ad libitum*. gBT-I mice encode transgenes expressing a T cell receptor recognising the HSV-1 glyco-protein B-derived epitope $gB_{498-505}$. OT-I mice encode transgenes expressing a T cell receptor recognising $OVA_{257-264}$ peptide. P14 mice encode transgenes expressing a T cell receptor recognising the LCMV glycoprotein$_{33-41}$ peptide.

## Flank scarification, HSV infection and TRITC dye painting

Mice were anesthetised with 1:1 ratio of Ketamine (Parnell Laboratories) and Xylazil (Troy Laboratories) (10 µg per g of body weight) via intraperitoneal injection. Moisturising eye gel (Alcon) was applied to the eyes to prevent dehydration. Mouse were shaven and the left flank and belly region were exposed. Veet cream (Reckitt Benckiser) was applied using a cotton bud to aid hair removal. Veet was removed using a wet tissue and dried using a dry tissue. Location for skin abrasion was marked with a marker pen just above the tip of the spleen on the lower left flank region. For experiments marking skin migrating cells, Tetramethylrhodamine (TRITC; Life Technologies) was dissolved in 2 µl Dimethyl sulphoxide (DMSO; Sigma–Aldrich) and subsequently diluted in 200 µl acetone (Chem Supply). Ten microliter of this mixture was applied between the marked abrasion area and the inguinal LN and allowed to dry for about 5 min.

Skin was lightly abrased by using a Dremel for 12 s to remove the epidermis. Ten microliter of $10^6$ pfu HSV was applied to the abrased skin region. A strip of the adhesive Opsite Flexigrid (Smith+Nephew) was attached to the infected area to contain the virus infection. Mice were bandaged with Micropore (3 M) and Transpore (3 M) and kept on a heating pad for about 2 h until recovered. Bandages were removed 2 days after infection.

## LCMV infections

LCMV virus was reconstituted at $2 \times 10^5$ pfu per 200 µl which were injected into mice intraperitoneally.

## T cell enrichment, labelling and adoptive transfer

To enrich naïve T cells, LNs (inguinal, popliteal, brachial, axial, mesenteric and liver) and spleen were harvested into phosphate-buffered saline (PBS) with 2.5% Foetal Calf Serum (Gibco) (PBS-2.5) on ice. Organs were mechanically disrupted by using a plunger from a 1 ml syringe as a mortar on a 70 µm cell strainer. Disrupted organs were topped up with PBS-2.5 and washed. For spleens, red blood cells were lysed with 1 ml red cell lysing buffer for 5 min and washed in 10 ml PBS-2.5. For further processing, spleen and LN suspensions were combined.

To enrich CD8+ gBT-I, OT-I or P14 T cells, the cell suspension was incubated on ice for 30 min in a CD8+ negative selection antibody mix (anti-erythrocytes [Ter119], anti-I-A/E [M5/114], anti-CD4 [GK1.5], anti-Gr1 [RB6-8C5], anti-Mac-1 [M1/70], anti-F4/80 [F4/80]). After incubation, cell suspensions were washed and resuspended in 6 ml (washed) BioMag Goat Anti-Rat IgG (Qiagen) for 20 min at 4 °C on an angled rotor. After incubation, CD8+ cells were enriched by running the cell suspension through a magnetic column. Enrichment purity was determined by flow cytometry with CD8α and Vα2 which typically ranged over 90%.

## Dye labelling and adoptive transfer

To label cells with CellTrace violet (CTV; Life Technologies) (5 µM stock), cells were resuspended at $10^6$/ml in PBS with 0.1% bovine serum albumin (BSA; Sigma–Aldrich). One microliter of CTV was added per

$10^7$ cells for up to 10 min in a 37 °C water bath. After incubation, stained cells were washed in PBS.

To label cells with CellTracker deep red (CTDR; Life Technologies) (1 mM stock), cells were resuspended at $10^6$/ml in PBS. CTDR was prepared 1:1 with diluted pluronic (Thermo Scientific; 1:10 in PBS). 1 µl of this prepared solution was added per $10^7$ cells and incubated for 15 min in a 37 °C water bath. After the first incubation, 10 ml PBS was added for a second round of incubation for 15 min in a 37 °C water bath. At the end of this last incubation, cells were washed in PBS.

Before adoptive transfer, viable cell number was determined by trypan blue staining on a haemocytometer. $0.5–1.0^6$ T cells were typically transferred intravenously in PBS.

## Confocal microscopy

Spleens were harvested directly into 2% Paraformaldehyde (PFA; ProSciTech) and fixed overnight at 4 °C on an angled rotor. The next day, fixing solution was removed by washing the spleens on a shaker in PBS for 5 min. After washing, spleens were transferred into a 30% sucrose solution and transferred to 4 °C for 24 h. After the tissues settled to the bottom of the tube, spleens were cut into three pieces (cross-sections) and frozen in OCT compound (Trajan Scientific) using liquid nitrogen and kept at −20 °C until needed.

Cryosections were cut at 25 µm on cryostat and air dried in dark for at least 10 min. Dried sections were fixed in acetone for 5 min and air-dried for 10 min. Each cross-section was marked with a square using a hydrophobic barrier pen before applying a drop of protein block for 10 min. Antibody mixes were prepared in 2.5% Normal Donkey Serum (NDS; Sigma–Aldrich) and left on ice in dark until needed. After protein block incubation, sections were cleaned from protein block solution (Agilent Technologies) and transferred to incubation chamber (Tupperware box with foil). Fifty microliter antibody solution was applied to each section and samples incubated overnight at 4 °C. Antibodies used: B220 (Pacific Blue, RA3-6B2, BioLegend, 1:200), CD3e (AF 700, eBio500A2, Thermo Fischer Scientific, 1:100), LCMV NP (AF 594, VL-4, Bio X Cell, 1:200) and CD11b (BV 421, M1/70, BioLegend, 1:200).

The next day, samples were washed twice in PBS for 5 min and mounted using ProLong Gold Antifade (Thermo Fisher Scientific). For spectral imaging, single colour controls were prepared, and images were acquired in lambda mode as per manufacturer instruction (Zeiss LSM 780) with a lambda bin width of 17.8 nm. Images were unmixed using ZEN blue (Zeiss, version 2.1) and further processed using Cecelia.

## Intravital two-photon microscopy

Inguinal LNs were imaged as described elsewhere[3]. In brief, mice were anaesthetised with isoflurane (Novachem; 2.5% to start and 1.0–1.5% to maintain and supplied with an 80:20 mixture of oxygen and air) and prepared for intravital imaging. If mice were not already shaved from a previous HSV flank infection, they shaved on their belly and left hind flank and remaining hair was removed using Veet. The first incision was made just right from the midline from the bottom of the belly to top of the ribcage through the dermis. The second incision was made following the vein of the left leg to the foot and further to the base of the tail. To disconnect the connective tissue, surgical scissors were used as weights while cutting–resulting in a 'tissue flap'. Vetbond tissue adhesive (3 M) was used to glue the 'tissue flap' dermis side down onto a stainless-steel platform. The inguinal LN was exposed by cutting away skin as a square window which was flooded with PBS to prevent dehydration.

Fat tissue was removed using micro scissors and gauze was utilised to remove eventual bleeding caused by small vessels within the fat tissue. Once the LN was cleared, a border of grease was applied to the square cut-out which could hold a pool of PBS. A square coverslip (ProSciTech) was placed on top, and a hydrophobic barrier pen (Sigma–Aldrich) was used to draw a square on top of the coverslip

where water could be placed for the water objective of the two-photon microscope.

Mice were placed in a custom build heated (35 °C) humidified chamber for two-photon imaging with an Olympus FV-MPERS multiphoton system and a 25× water objective. The system was equipped with two lasers Mai-Tai and Insight DeepSee which were used at 820 nm (for CTV) and 980 nm (for GCAMP, CTDR and or TRITC) wavelengths respectively. The time interval between frames was 10 s, with a 5 μm gap between optical planes. Images were processed using Cecelia.

## Statistics and reproducibility

No statistical method was used to predetermine sample size. Images with low quality, high autofluorescence or uncorrectable tissue drift were excluded from the analysis. The experiments were not randomised. The Investigators were not blinded to allocation during experiments and outcome assessment.

## Image analysis

An overview of packages used in Cecelia is listed in Supplementary table 1. For tutorials and user interface interaction, the reader is referred to https://cecelia.readthedocs.io. Images were imported as OME-Zarr[15] using bioformats2raw (Glencoe Software). For Xenium files, point data was converted into images with a local sum of pixels to aggregate and make transcripts easier to view and analyse.

Images underwent 'cleanup' routines if needed to account for overspill between individual channels. This is especially important for two-photon images as they contain, generally, a significant amount of autofluorescence and channel spill-over. While correction in Imaris is often done using channel subtraction, we opted for channel division instead. Channel subtraction often requires a multiplication factor to account for intensity differences between channels. This factor might vary from image to image. Our channel division method did not require individual adjustments across images which enabled batch processing. This cleanup step can be further improved with the recent Cellpose denoising framework[37] to account for intensity inhomogeneity within imaging planes.

Subsequently, images underwent cell and structure segmentation. While we integrated various deep learning approaches into the framework, Cellpose[16] has proven to be the most practical as it is widely applicable to various cases and custom models can be generated relatively simply. Depending on whether the image is static or live the identified labels undergo population definition or tracking respectively.

Populations were defined by flow cytometry gating, implemented using flowWorkspace, or by population clustering, using Leiden clustering. For tracking, we utilised the Bayesian tracker btrack as it provided convenient integration into Cecelia. We currently have no option to manually correct the resulting tracks. Tracking statistics were computed with celltrackR. Populations and tracked cells were further utilised for spatial analysis.

As the number of cells in static images can be rather large, most analysis was done based on centroids with packages such as DBSCAN. For two-photon analysis, 3D meshes were generated using trimesh which also provides a collision manager to detect cell interactions. For dynamic cell behaviour, the HMM functionality from depmixS4 was utilised and combined with Leiden clustering. Data plots were generated in R or GraphPad Prism 10 (GraphPad Software).

## Development of Cecelia

Image analysis is often a combination and integration of different tools, packages and frameworks from various programming and scripting languages. This diversity can increase ease of use for specific scenarios. The downside of this approach is a proliferation of packages that researchers must traverse to analyse their images. We, therefore, aimed to develop a toolkit that would support research questions specific to the immunological research in our lab but would also support cell biology research in general.

Python has been adopted by many image analysts due to its programming accessibility, vast array of processing libraries and relative ease for non-programmers to obtain reasonable automation of results. The recent development of a multidimensional image viewer in Python, termed napari[14], has expanded this accessibility to image processing and opened up the possibility to implement custom workflows relatively fast within a simple user interface. Within napari, users have the capacity to develop custom plugins for specific or more generic applications which resulted in numerous custom plugins in a relatively short time (https://www.napari-hub.org/). With the advent of the multi-core and parallel computing framework Dask (https://www.dask.org/) and the Next Generation File Format (NGFF) from the Open Microscopy Environment (OME)[15], new and more accessible options are now available to parallelise processing steps and to store image information in an open and extensible community supported image file format. Downstream of image processing, respective data analysis is dependent on expertise in the research group. Popular choices include Python, MATLAB and R. Python and R have similar analysis capabilities and are in many cases interchangeable.

The question in this context is, whether and how these tools could be integrated into a generic toolbox for cell biologists to conduct image processing and downstream analysis within the same workflow. We opted for napari as the main image viewer because of its increasing popularity and sustained support by the Chan Zuckerberg Initiative (https://chanzuckerberg.com/science/programs-resources/imaging/). The downside of napari is that it must run locally. With the advent of more web frameworks and remote cloud computing, there are a variety of web frameworks available to view and analyse images[38,39]; however, we could not find a suitable image viewer that runs in a web environment and supports 2D, 3D and 3D+time images with the option to display images larger than commonly available local memory and cell tracks.

Napari alone however might not be suitable for an extensive framework to process, manage and analyse imaging data as its primary use is viewing images. We therefore considered several options to combine image processing and viewing using napari with the ability to interactively generate data plots and images together with downstream analysis—these include: a native napari plugin with plotting engine, napari embedded into another Python Qt-based GUI and napari in combination with a reactive web-framework. The option to add use case-specific plugins to napari is very powerful. Images can be directly linked to simple plots, such as clustering segmented objects and viewing them interactively (https://github.com/BiAPoL/napari-clusters-plotter). While these approaches will elevate how researchers can work and interact with images, our aim was to provide a comprehensive toolbox that supports different kinds of plots and tables in an interactive manner with the possibility to extend this as different use cases arise in the future without the need to reprogramme visual aids but rather to make use of the variety of packages available in any scripting or programming language.

As napari is a native Qt application (a cross-platform framework to create user interfaces, https://www.qt.io), it could also be embedded into a another Qt-based application as done for the application PartSeg which was designed to segment structures within nuclei[40]. This solution would add more interactivity around napari but would also require the implementation of a significant amount of interactive plotting functionality which is currently available within Matplotlib (https://matplotlib.org), albeit limited. Reactive web frameworks such as Dash (Python) and Shiny (R) have significantly reduced the barrier to display data interactively with a reduced amount of implementation work required. Dash and Shiny have similar functionalities and utilise the same plotting engines such as Plotly (https://plotly.com). Shiny's reactive programming capability can reduce the amount of code

required to produce the same output while also enabling to store intermediate variables which is not possible in Dash. R further has the advantage of having a strong bias towards statistical analysis packages and tests. Keeping data analysis and image viewing in separate 'compartments' therefore opens up the space to benefit from both approaches, that is, napari's vast plugin system and the wide variety of interactive libraries available for reactive web frameworks.

## Design and implementation of Cecelia

Cecelia consists of a user interface in R/Shiny and Python/napari (Supplementary Fig. 1a). The main user interface for image, task and population management was implemented in R while image processing, segmentation and other related functions were implemented in Python. The main components are structured into (1) module pages, (2) napari viewer managements, (3) project management, (4) image management, (5) task management and (6) population management (Supplementary Fig. 1b). The user mainly interacts with the module pages from Shiny while the individual tasks can be provided by custom modules which are implemented in R and/or Python depending on the functionality needed.

A Jupyter kernel serves as the communication platform between Shiny and napari. The way back from Python to R is utilising JSON files that are written by napari and read into Shiny using a *reactivePoll*, for example when the user selects points on the image during gating to show them on the gating plot. For image processing, we utilised the recently developed Open Microscopy Environment Next Generation File Format (OME-NGFF)[15] which is based on small compressed files in combination with Dask for lazy loading and processing of large-scale images. Every image is represented as a reactive persistent object (RPO) within R. 'Reactive' means that the object acts as any other reactive variable within Shiny. This design was based on a theoretical description within the R/Shiny community (https://community.rstudio.com/t/good-way-to-create-a-reactive-aware-r6-class/84890/8). 'Persistent' means that the information of every object is saved to individual RDS files. Every object is therefore saved independently and can be moved between computing environments to process data in different physical locations. For all processing steps, especially deep learning (DL) based segmentation, the user has the option to run every operation on a High-Performance Computing (HPC) system with the advantage of increasing computation, memory and graphics card resources.

For each unique RPO, there is a corresponding folder with its unique identifier to store all object-related data and analysis. Every project, which also has a unique identifier, holds corresponding experimental sets with individual images. A project can contain a series of experiments that are related to each other and require similar sets of analysis workflows. Projects are stored at the root of the folder structure. Each project has several versions, where version '0' contains all image information and versions '1' to 'n' contain all the processed information. Versions can be created by the user as a backup to test whether the experimental data could be analysed differently by saving a specific state of the already analysed data. Within each data folder, there are several subfolders that contain segmentation labels and properties as well the population definitions.

## Task management in Cecelia

Each RPO can be used to process tasks. A major drawback of using R for process management is that there is no support for concurrent threads. Individual tasks can however run in separate parallel processes. Within Shiny, there is currently no direct module or library that manages processes. Our aim was to provide a framework that could utilise R and Python packages within tasks that can run locally as well as remotely on HPC systems. The current implementation works with the R package mcparallel to run individual forked processes; an approach which only works on Unix systems.

The current implementation is based on two main classes: *taskLauncher* and *taskProcess*. *taskLauncher* is a task container which provides functions to start, stop and retrieve results. All tasks inherit from *taskProcess* which provides functions to retrieve the given parameters and work with the RPO of the current task. Generally, each task is coupled to an RPO where logfiles and results will be written to. When multiple images are being processed, the experimental set of these images will be taken as the processing RPO. For tasks running on the HPC, the current RPO state file (RDS file) will be uploaded together with the task parameters. The HPC job will start an R script to run the task.

Processing within Python is done by calling corresponding Python scripts in R via system calls. Certain features, especially Dask, are currently not working when using the R package reticulate which is used to run Python libraries within R, hence Python scripts are called directly rather than integrated within R. Python tasks are handled like R tasks in the sense that parameters are passed to the script via JSON files.

Every task function has a unique name across the app; for example, Cellpose segmentation is called *segment.cellpose*. This name is used to link functions to individual module pages within Shiny; that means, all tasks starting with *segment* will be shown on the segmentation module page. Processing parameters for every function are defined in JSON files. Based on the input definitions within these JSON files, the input parameters to be shown in the Shiny app for every function are generated and then passed to the corresponding function call. Parameters passed in this way are then available in every process by calling the function *self$funParams()*. In this way, it is relatively simple to introduce and implement new functionality within the app. The developer must define a JSON file for parameters and an R file for the function implementation. Commonly used functionality such as adding populations, running tasks or plotting data in flow cytometry plots is also implemented in this modular way. The app is therefore extendible beyond the currently available modules.

## Installation options

We provide detailed instructions for installation of Cecelia online: https://cecelia.readthedocs.io/en/latest/installation.html.

Windows/Docker Installation: The Docker solution (independent of operating system) is the easiest to install. However, our Dockerfile is based on 'rocker' which is based on linux/amd64. This means that on Apple Silicon-based machines the container will run in emulation mode which significantly decreases the app performance. We therefore recommend installing the native R-package whenever possible for performance issues.

MacOS installation: R-package installations can vary vastly between systems. We, therefore, opted to use 'renv' to freeze package versions in R and 'pip freeze' to snapshot Python libraries. Although this means that the packages/libraries will not always be the most recent versions, there should be no version conflicts that might cause issues for users. Note that as we are not actively developing with Windows, we have limited knowledge of how well this approach will work on Windows systems.

Linux installation: as for MacOS, we have opted to utilise 'renv' to create a more reproducible environment.

Manual package installation: For more advanced users, installation of the application and associated packages is possible without using 'renv'. This option should be for people who want to contribute to the package development.

## Running Cecelia with Docker

The app utilises a wide range of packages, libraries and several external programs. We have aimed to make the installation as minimal as possible by providing an R package structure with integrated Python scripts. The user then must run a series of initial methods to install

further package dependencies and retrieve DL models. For ease of usage, we provided a contained running environment using a Docker container. It is currently not easy to get napari working within a Docker environment. We have therefore chosen to install napari within a conda environment on the host. This environment communicates with Shiny which is packaged within the Docker container. This setup is currently required for Windows users because packages such as the 'parallel' package and its 'mcparallel' function are not available for Windows systems. We further use other command-line tools that are common for Unix systems to work with files which are not native to Windows systems. Future versions will seek to expand on native Windows support.

## Image segmentation and object measurement

DL-based methods are arguably the most efficient and effective means for image segmentation[16,41,42]. Whole-slide 2D images with nuclei and membrane staining can now be relatively reliably segmented and analysed even in crowded conditions. Our ambition was to provide workflows that could also segment larger images in 3D and 3D+time with varying cell sizes and shapes without the user having to retrain models. We have therefore incorporated the commonly used DL frameworks Mesmer[41], Stardist[42] and Cellpose[16]. For our purposes, Cellpose has been the most useful as it can segment a variety of cell shapes and sizes in 2D and 3D without the need to create a series of custom models.

After cell segmentation, objects must be measured for further analysis. Mesmer[41] and a recent publication on two-photon image analysis[13] provided a wide variety of measurements for 2D and 3D object analysis respectively. These 3D measurements required the generation of surface meshes. There are several libraries available within Python to create and manage meshes, we opted for trimesh (https://trimsh.org/). Trimesh can extract meshes from binary images via scikit-image's marching cube algorithm. These meshes can be saved for later use to detect object interactions which trimesh supports via a Collision Manager. We have currently not implemented a way of combining meshes into single object files which results in every mesh being saved as a single STL file (a file format to store vertices) which takes up a significant amount of storage space for larger images and results in slow processing as meshes must be loaded individually from disk.

## Object storage and access in Cecelia

There are several options available to store data. We are using R and Python together which required an approach to read and write data from both sides. We opted for the HDF5 based file-format Anndata (https://github.com/scverse/anndata) which was originally developed for single-cell analysis within the Python package Scanpy[43]. The data can be accessed from R via the reticulate library. For each segmentation, object measurements and properties are saved within a single Anndata file. For easier access, we implemented a helper class called *LabelPropsView* which provides functions for the most used access options. One challenge is that R and Python do not operate in the same process and therefore access to these data files can lead to a 'race condition' where both environments might access the same file at the same time. Operations between these two systems therefore must be executed in a coordinated manner.

## Population management in Cecelia

We aimed to support different kinds of imaging data including tracked, gated and clustered cell populations. As each of these has slightly different requirements, we decided to split these into different classes with one common base class. Currently, all populations are defined and saved within R and accessible in Python as read-only. This decision was made initially as the gated cell population class is based on the R package flowWorkspace which provides a population gating framework. The population IDs for every cell population are written out as individual CSV files that Python can read out and filter the main dataset accordingly. Other sub-populations from tracking and clustering are implemented as 'filtered' populations; for example, tracked cells have a *track_id* above 0 which is used to define the 'tracked' cell population. These filters are again applied within R which is then saving the population IDs to CSV files for Python to display. Clustered populations and regions work in the same manner.

## Population gating in Cecelia

One of the main functionalities that were required for the app was to be able to gate on cell populations and check the gating result at the same time on the image. Our approach was to try to utilise existing components from Shiny and R to create a gating interface. Plotly (https://plotly.com/) is one of the most popular frameworks for interactive graphing and plotting which is integrated into Shiny and Python's dash (https://plotly.com/dash/). Plotly enables drawing of shapes onto the data which can be utilised for creating population gates. Every time the user creates a shape, a *relayout* event is triggered by the plot. This *relayout* event contains the points data from the drawn shape which can be used to create or change the population gating.

Gating of cell populations is essentially a simple 2D shape which is drawn on top of data points. flowWorkspace is an R package that was designed to represent gating hierarchies and import and export data from and to the commercial and commonly used FlowJo software (https://www.flowjo.com/). The combination of these two packages, Plotly and flowWorkspace, together with Shiny's reactive framework enables interactive population gating. The current implementation is however comparatively inefficient due to the nature of reactivity. Future developments should more carefully consider the dependencies and interactions of reactive variables and respective functions and whether plotly could be replaced by a custom-designed plotting library for flow cytometry, as done in OMIQ, to increase efficiency and user experience.

One issue for displaying segmentation data is the large number of points with larger images. A commonly used method is to use raster images instead of plotting individual points. We utilised the R package rasterly (https://github.com/plotly/rasterly) to generate raster plots and contour plots to achieve this.

## Cell tracking in Cecelia

Object tracking has been widely simplified with the advent of the ImageJ plugin Trackmate[44] which was recently expanded to also track results from DL segmentation frameworks[45]. Here we utilised the Bayesian tracking package btrack within Python[28] to track results from segmentations.

## Neighbour detection in Cecelia

Neighbour detection can be done in several ways, such as a fixed scanning window across the image, cell-centred radii or Delaunay triangulation. Squidpy[46] was recently developed to analyse spatial single-cell data and is based on Anndata. We can therefore utilise the functions therein to obtain cell neighbours using the mentioned methods.

## Plotting data in Cecelia

One advantage of R/Shiny is that generating data plots is relatively straightforward. Shiny will render any plot that can be produced within R as image files that are then displayed on the web application. Processing and analysing imaging data has several manual and comparatively subjective steps where users must choose parameters that will influence the downstream analysis and the final figure. We therefore aimed to provide a workflow where the effects of certain parameter choices can be examined interactively. To display data in different ways and interactions, we adopted a series

of previously defined Shiny apps (https://huygens.science.uva.nl/) and integrated them into Cecelia within the 'Plot Canvas' section. The data for the plots is taken from the analysed images which results in direct visualisation of the processed images so that users can see the impact of different processing parameters on the image as well as on the resulting data plot.

## Reporting summary

Further information on research design is available in the Nature Portfolio Reporting Summary linked to this article.

## Data availability

The imaging data generated in this study have been deposited in the Zenodo database under accession code https://doi.org/10.5281/zenodo.14759798. Images not generated in this article were downloaded from the sources outlined in Supplementary Table 2. Source data are provided with this paper.

## Code availability

Code can be accessed at https://github.com/schienstockd/cecelia[47].

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

## Acknowledgements
This research was supported by The University of Melbourne's Research Computing Services and the Petascale Campus Initiative. This work was supported by a National Health and Medical Research Council Investigator Fellowship to S.N.M. (2017220). We thank the Biological Optical Microscopy Platform at the University of Melbourne for access to microscope facilities and the Biological Research Facility for management, breeding and maintenance of mice. We thank Zoe Fransos, Ziwei Luo, Harry Horsnell, Justine Seow, Ali Zaid, Tri Phan, Xufeng Lin, Deborah Barkauskas, Kes Barkauskas, Luis Alarcon-Martinez, Edwin Hawkins, Jeremy Er, and Shihan Li for help testing Cecelia.

## Author contributions
D.S. and S.N.M. conceived the project. D.S. performed experiments, data analysis, computational implementation and software development. J.L.H. and S.D. provided experimental data and provided feedback on the manuscript. D.S. prepared the manuscript and D.S. and S.N.M. edited the manuscript. S.N.M. supervised the entire project.

## Competing interests
The authors declare no competing interests.
