## [Transparent Peer Review file · Nature Communications]

Cecelia: a multifunctional image analysis toolbox for decoding spatial cellular interactions and behaviour

Corresponding Author: Professor Scott Mueller

Version 0:

Reviewer comments:

Reviewer #1

(Remarks to the Author)

The work is indeed contributive to the community. The manuscript is fine with a few minor issues. However, I would recommend a major revision over Cecelia installation resources and GitHub documentation for poor accessibility. Recommend future publication upon problems solved. Details below:

The authors created Cecelia as an integrative platform to conduct multi-purpose quantitative analysis over 2D/3D images or videos. Their toolbox (theoretically) will provide a significant leap over the accessibility and efficiency over analyzing images describing intercellular patterning and interaction. By providing solid examples (Fig. 2-5), the authors proved that the modules that the authors transplanted or invented can track, measure, describe, and cluster the static and dynamic features of cell groups, enabling easier identification of features of certain cell groups and their interaction. On the other hand, customized algorithms and libraries are able to be applied or transplanted to Cecelia (which also serves as a universal data management and processing platform), which further appreciatively enable this app to contribute to a far broader range our community. Beyond the substantial content of Cecelia, most part of the text of this manuscript is concise and clear, except for a few minor issues of lacking reasonable details and misleading words (described herein later). From those perspectives, I believe the research and manuscript per se are fine and without major concerns.

However, I believe authors did a bad job in providing enough accessibility for different types of users (or perhaps any users not experienced in app development) because while they allegedly provided accessibility to Mac, Linux, and Windows (docker only), there are bugs, logic flaws, and insufficient information which prohibit users from installing Cecelia. I am a cell biologist with certain level knowledge of bioinformatics and image processing, who has developed and maintained python library but not complete app like the authors, and I believe I may represent user with intermediate level of coding and developing among all potential users. However, I have been trying to install this app for 6 hours but finally decided to commit that I failed. Below is what I found:

For windows (Docker, <https://github.com/schienstockd/ceceliaDocker>):

I am a windows user so I first tried this one, but I get stuck at "conda env create --file=conda-gui-env.yml", because "conda-gui-env.yml" not found. Where is this document? I cannot create Conda environment if the authors don't tell me where this yml is. Even if I could pass this line, I did not see any description over where other .yml and .bat files are, so I am completely lost how to install it. Also, the authors claim this is a docker version, but they didn't mention anything about how to use it in Windows docker desktop app, which I thought is the most general way to use docker. I would like to learn how to make Cecelia available in docker desktop, which is supposed to be a more user-friendly way.

For Linux (<https://github.com/schienstockd/cecelia>):

First, the authors failed to claim they directly start in R environments. They have to say it in the documentation otherwise users without coding experience may not be able to run any line. Second, the authors failed to claim what is the requirement of R version. I am using Windows Subsystem for Linux, Ubuntu 22, to test this, but when I downloaded the default version of R and up-to-date version of R studio server edition and run this line

```
"remotes::install_github("schienstockd/cecelia", Ncpus = 4, repos = "https://cloud.r-project.org")"
```

to install, it turns out error because a few packages are not able to install due to conflicted versions.

For "Macs" (<https://github.com/schienstockd/cecelia>):

I am actually able to follow their Macs installation approach on my Windows. It is somehow interesting because the authors claimed "This package currently only works on Unix systems because shiny requires this architecture", and therefore

Windows version is only accessible through docker. However, I search the Shiny documentation and asked ChatGPT, surprisingly found that Shiny should be able to run on Windows without any problem. So I directly tried to run this process on my Windows, and did not find any problems related to Shiny. What really stopped me is an error because windows does not support a python library "glmnet" when I was running "cciaCondaCreate(envType = "image")". However, I think this is avoidable, because (1) it is a non-maintained library so the authors may swap it; (2) its function can be altered by other libraries; and (3) R has the same library that is actually supported by windows so you may use it instead. Therefore, I strongly require authors to check again whether Windows are able to use Shiny and hence Cecelia. If Shiny is indeed available, there should be an easier way for us Windows users to install it, either through Dockers or Bash or R command lines.

Therefore, the authors need to carefully inspect and edit their GitHub documentation as well as the app deployment and installation, and finally provide easy and accessible approaches to install Cecelia for potential users with different background and devices. While I certainly believe Celelia is a strong and impactful toolbox that deserves to be published in Nature Communications, I cannot support the publication with the current documentation until the authors make the installation clear with necessary details, without bugs and misleading info, and accessible to all types of users. While this may be regarded as a major concern, I believe it is not difficult to solve. I would recommend editors to learn from authors if they are able/willing to handle the accessibility issues, to determine the editorial decision.

There are some minor issues listed below for authors to consider:

Line 39-42 "current approaches ... each step.": Lack of detail. You would like to add information describing how complicated and laborious the previous approaches are. For example, what is a typical classic approach pipeline like? How many hours of user's effort does it take to finish the script and process one sample?

Line 78-80 "Furthermore ... Python notebooks." This is extremely good and necessary, but you need to provide detailed explanation in your website documentation about how to enable this. For example, how to call data from your pipeline? How to make customized module and implant into Cecelia processing pipeline?

Line 99 "Cellpose ... (cyto2)". Please provide a reference for this model.

Line 98-102 "Segmentation ... analysis.". Unclear/confounding concept? What do you mean by "segmentation" here? In my dictionary, "segment" in the field of image processing means one connected area of pixel/voxel labeled with certain identity in the image. In this case, it means each identified cell. Accordingly, "segmentation" means process to identify and label individual cells in each channel. If this is true, then what do you mean by "segmented channels were then merged into a combined segmentation"? I think "segmented" channels means some type of cell labeling; but if you combine them together, should they be called "segmentation" anymore? It would be just a total labeling dataset. Or do you mean something totally different from what I am thinking about? Please specify.

Line 112. What is the full name of IBEX? Also please provide a very brief introduction, perhaps plus ref.

Line 117. Please provide a brief introduction of Leiden clustering because not all readers understand this concept. Perhaps also provide reference. Correspondingly, in Fig. 3A, please describe in the legend what x-axis means in your Leiden clustering.

Line 125. Same issue. What is the full name of CODEX? Also please provide a very brief introduction, perhaps plus ref.

Line 130. Ref for Delaunay triangulation?

Line 125. Same issue. Please provide a very brief introduction plus ref for Xenium in situ analysis.

Line 151-155. Please provide a very brief biological background for people without corresponding immunological knowledge.

Line 151. Strange sentence. An easiest way is to change "imaged" to "found".

Line 164. Recommend a ref for HMM.

Line 166. Will you consider changing the terminology "angle"? This name does not reflect the concept you are measure. If my understanding is correct, you are measuring the cumulative angle of change of direction of the cell, but generally when general reader hear "angle", we cannot reflect on cell movement there is no absolute coordinate system and cell always change direction. Therefore I highly recommend you change this terminology to "cumulative change of direction" or whatever similar.

Line 190. What do you mean by "segmentation" here? I don't understand.

Fig. 2-5. For many microscopic images you labeled in the top left with 3 colored circles, does that mean the total fluorescent dye or tested mRNA number you involve, i.e., channel number? Whatever it means, please make sure you indeed describe it in the legend. Also, for those 2D/3D labels, I strongly recommend you change them to x-y, x-y-z, x-y-t, or x-y-z-t, because readers need to easily understand the data structure you are using. When people see "3D", they don't understand if you are talking about voxel image or video.

Line 199-201. Please provide a very brief biological background to let readers understand what you are doing. Please also mention the word "silence", so that more general readers can easily reflect that shRNA is an approach to silence target gene in your mammalian cell system.

Line 202. It is a good idea to apply ILEE anisotropy function in this way, but when you were using ILEE anisotropy function, did you adjust tensor box size? ILEE is designed for cytoskeleton in a single cell, whose tensor box size by voxel may not apply to your usage looking at large amount of cell group. So, you may need to adjust to make your measurement accurate and reflecting your feature appropriately.

Fig. 2B. Please use the same field of view for each image, or mark which part in the merged image you used to show the split channel.

Fig. 4A. Image presentation is misleading. I think you need to mark on panel A that this is x-y-t or x-y-z-t video and also state in the legend that this representative frame of your video which is not looking the same area of the same sample at different time point.

(Remarks on code availability)

I spent 6 hours and failed to install it. Overall accessibility is their only major issue, so I wrote remarks over the app in the main comment, as above. Please check.

Reviewer #2

(Remarks to the Author)

In this manuscript, Schienstock et al present a new open-source image analysis toolkit, Cecelia, that integrates multiple existing pipelines into a unified platform. In addition to advanced analysis of static images, Cecelia provides the ability for interrogation of time-resolved data, such as by incorporating dynamics of cell migration with Hidden Markov Models to allow advanced classification of cell behavior with respect to tissue location. Cecelia is not entirely novel in itself, as it is an integration of previously developed algorithms and pipelines. However, this integration is important as it generates a tool and resource for the bioimaging community which will help with standardized quantification of diverse imaging datasets.

The manuscript is generally well-written. Detailed instructions for installation and use are also provided in the supplementary material and on a separate website, making this platform relatively easy to use/test.

The main concern is regarding the level of detail provided to the reported analyses and results. Most figures and accompanying text are very general and lack sufficient detail to understand the specifics of how the results were generated or what exactly is shown. Here are some specific areas that could use improvement in reporting:

- Figure 2D: How were clustered vs non-clustered T cells defined? What is meant by the term "in contact with"?
- Figure 3A: How were B cell rich, T cell rich, and Macrophages regions defined (manual vs algorithmic)?
- Figure 3B: is the UMPA clustering showing cell clustering or region clustering? For the spatial image of tissue regions, is it cells or regions which are shown? If cells, then this is not analysis of tissue regions, rather of cell type distribution. As above, if the data show cell type clustering, then how were the regions identified?
- Figure 3C: same as in 3B - what is shown in each panel, cells or regions? How were region interactions quantified?
- Figure 4H-K: It would be helpful to describe in greater detail the workflow and results of these panels. As currently written, the reader has to guess what is shown, how things were calculated, and what the graphs demonstrate.
- Figure 5A: What was the cutoff used for determining if cells are close to versus away from lymphatics? It would be good to show two separate UMAP plots or color-coded dots for cells close to and away from the lymphatics to demonstrate how these differ. How does this compare to standard velocity measures?
- Figure 5B: The text states that MYO10 targeted samples had decreased migration velocity. It is not clear where these data are shown and whether these results are significant. How does this compare to standard velocity measures?

(Remarks on code availability)

Reviewer #3

(Remarks to the Author)

In this manuscript, the authors present Cecelia, an innovative open-source toolbox designed to integrate advanced quantitative analysis of both static and live cell imaging. The toolbox provides a unified framework for multidimensional image analysis, enabling researchers to handle complex imaging datasets and conduct spatial and behavioral cell analysis with well efficiency. By incorporating Hidden Markov Models (HMM) and real-time image processing with a user-friendly interface, Cecelia offers a powerful solution for studying dynamic immune cell behaviors and interactions. This work significantly contributes to the field of biological imaging by addressing the challenges of analyzing highly multiplexed datasets. However, for a more complete manuscript, we present some supplements and corrections.

Minor Comment

1. In the Introduction section (p.2), it would be helpful to add a brief explanation to clarify the statement "Studies have highlighted that the 3D tissue organisation is critical to understand immune responses" for better reader understanding.
2. Similarly, to facilitate the readers' comprehension, it is necessary to provide an explanation and references in the sentence "Current approaches rely on a sequence of software tools ~" to clarify what specific issues arise from this approach.
3. In the Methods section, a brief introduction and references to the methods used should be added. For example, methods like Leiden clustering and KMeans clustering should be briefly explained and referenced.
4. In the Discussion section, it would be beneficial to include a description of Cecelia's limitations and potential future developments.
5. In the Result section(p.3), The author mentioned that Cecelia, which was developed, is capable of real-time analysis of resulting data on microscopy images. Does the analysis time vary as the number of cells observed increases? It would be good to add the results of the analysis on the execution time of the cell tracking algorithm.
6. In the Methods section(p.27), Before performing cell and structure segmentation, a cleanup process was applied to the

images, and it is mentioned that this resulted in a cleaner output compared to commercial software such as Imaris. However, there are no comparison results provided to support this claim. Including such a comparison would make Cecelia's advantages more convincingly presented.

7. Is there a function that allows for manual correction of errors during the cell tracking process? How do you classify cells with errors in the tracking results?

(Remarks on code availability)

Version 1:

Reviewer comments:

Reviewer #1

(Remarks to the Author)

Same as previous opinion, the paper should significantly contribute to related scientific community that demands powerful and accessible image computation toolbox. Below is my updated opinion on the current renewed submission:

1. All minor issues are fixed in the main text and figure, at least to a level that I don't have concern.

2. They indeed rewrote the GitHub documentation. Significant improvement. Now I am finally able to successfully install Cecelia to my Windows platform using Docker. However, I still spend 2 hours to debug installing as there are still a few flaws, or misleading content, while some are not necessarily attributable to the authors:

(1) Miniconda and Anaconda are not identical. However, official page of "install Miniconda" leads users to actually install Anaconda, which generates a huge problem here. Both work, but their default environment PATH folder is different, and the users might misunderstand as they indeed installed Miniconda, using default setting. But the authors say in their documentation:

"If you installed Miniconda in a custom location, ie/ not your user account, you must specify that directory in the .command or .bat file by editing the file in a Text editor."

If users don't know what is going on, they will think they are using default Miniconda PATH and do not change anything. But the real PATH for Anaconda, which they are directed to install, is:

C:\Users\xxx(my user name)\anaconda3

Therefore, the install .bat will NEVER work. So you should provide additional instruction that following the "install Miniconda" page leads to Anaconda but not Miniconda, and provide instruction how to change their CONDA_DIR.

(2) They failed to tell that when you run cecelia-Windows-docker.bat, you must have Windows Docker app opened. Again, not all ppl have used independent Docker pack outside Docker app; also, not all ppl have used Docker by any means. If I didn't feed their total documentation and my bug report to GPT-o1. I would never believe I waste more than 1 hour on such a ridiculous issue.

3. Since I conduct different type of research, I don't have appropriate sample to test performance. But the Cecelia app build quality looks high.

Although there are still concerns on GitHub Documentation, I agree to publish because Cecelia is indeed accessible now, with problem solved. However, I recommend the authors edit their GitHub to solve the problems sooner than later.

(Remarks on code availability)

I didn't review detailed code, but checked their installation documentation carefully. Comment as above.

Reviewer #2

(Remarks to the Author)

The authors have adequately addressed all the raised concerns, and I have no further comments. Congratulations on an excellent paper and platform.

(Remarks on code availability)

Reviewer #3

(Remarks to the Author)

The authors have addressed the reviewer's comments effectively, and the revisions have substantially improved the overall quality of the paper. I believe the manuscript is now suitable for publication.

(Remarks on code availability)

The code is designed to run on a Docker-based environment, and the program is built using an R-based language. It is enough to reproduce the results of the paper.

Point-by-point response to Reviewers

Reviewer #1 (Remarks to the Author):

The work is indeed contributive to the community. The manuscript is fine with a few minor issues. However, I would recommend a major revision over Cecelia installation resources and GitHub documentation for poor accessibility. Recommend future publication upon problems solved. Details below:

The authors created Cecelia as an integrative platform to conduct multi-purpose quantitative analysis over 2D/3D images or videos. Their toolbox (theoretically) will provide a significant leap over the accessibility and efficiency over analyzing images describing intercellular patterning and interaction. By providing solid examples (Fig. 2-5), the authors proved that the modules that the authors transplanted or invented can track, measure, describe, and cluster the static and dynamic features of cell groups, enabling easier identification of features of certain cell groups and their interaction. On the other hand, customized algorithms and libraries are able to be applied or transplanted to Cecelia (which also serves as a universal data management and processing platform), which further appreciatively enable this app to contribute to a far broader range our community. Beyond the substantial content of Cecelia, most part of the text of this manuscript is concise and clear, except for a few minor issues of lacking reasonable details and misleading words (described herein later). From those perspectives, I believe the research and manuscript per se are fine and without major concerns.

However, I believe authors did a bad job in providing enough accessibility for different types of users (or perhaps any users not experienced in app development) because while they allegedly provided accessibility to Mac, Linux, and Windows (docker only), there are bugs, logic flaws, and insufficient information which prohibit users from installing Cecelia. I am a cell biologist with certain level knowledge of bioinformatics and image processing, who has developed and maintained python library but not complete app like the authors, and I believe I may represent user with intermediate level of coding and developing among all potential users. However, I have been trying to install this app for 6 hours but finally decided to commit that I failed. Below is what I found:

We thank the Reviewer for their supportive and constructive feedback, and we apologise for not making the installation instructions as clear as we had intended. We have now substantially revised and independently tested the installation instructions for Mac and Windows operating systems to make the process more intuitive for users a broader range of expertise working with software: <https://cecelia.readthedocs.io/en/latest/installation.html>

Furthermore, we have added a detailed overview of the app components to introduce new users to the Shiny application and the napari viewer: <https://cecelia.readthedocs.io/en/latest/overview.html>.

Additionally, we have made a series of walkthrough tutorial videos to help new users become familiar with the location of features and how the core functions operate before attempting the

more in-depth tutorials and examples workflows, which are also provided on the same webpage: <https://cecilia.readthedocs.io/en/latest/walkthrough.html>

We understand that the choice between the different installation options could be confusing, and we should be clearer about which solution is the best fit for users. We provide detailed updated instructions for 4 installation options in the Methods section of the manuscript:

“Installation options

We provide detailed instructions for installation of Cecelia online: <https://cecilia.readthedocs.io/en/latest/installation.html>.

Windows/Docker Installation: The Docker solution (independent of operating system) is the easiest to install. However, our Dockerfile is based on ‘rocker’ which is based on linux/amd64. This means that on Apple Silicon based machines the container will run in emulation mode which significantly decreases the app performance. We therefore recommend installing the native R-package whenever possible for performance issues.

MacOS installation: R-package installations can vary vastly between systems. We therefore opted to use ‘renv’ to freeze package versions in R and ‘pip freeze’ to snapshot python libraries. Although this means that the packages/libraries will not always be the most recent versions, there should be no version conflicts that might cause issues for users. We have tested this with different users. Issues that can arise relate to incorrectly installed C++ compilers and Xcode on the user side. We have provided examples of these errors and clear instructions for resolving such issues. Note that as we are not actively developing with Windows, we have limited knowledge how well this approach will work on Windows systems.

Linux installation: As for MacOS, we have opted to utilise ‘renv’ to create a more reproducible environment.

Manual package installation: For more advanced users, installation of the application and associated packages is possible without using ‘renv’. This option should be for people who want to contribute to the package development.”

For windows (Docker, <https://github.com/schienstockd/ceciliaDocker>):

I am a windows user so I first tried this one, but I get stuck at “conda env create --file=conda-gui-env.yml”, because “conda-gui-env.yml” not found. Where is this document? I cannot create Conda environment if the authors don’t tell me where this yml is. Even if I could pass this line, I did not see any description over where other .yml and .bat files are, so I am completely lost how to install it. Also, the authors claim this is a docker version, but they didn’t mention anything about how to use it in Windows docker desktop app, which I thought is the most general way to use docker. I would like to learn how to make Cecelia available in docker desktop, which is supposed to be a more user-friendly way.

We apologise if the location of some of the files was unclear. These files are contained within the “ceciliaDocker” folder that is downloaded from Github during the install process. The ‘.bat’ file will start the jupyter server and the docker image. The user should then be able to just run this ‘.bat’ file for the app to start. Docker desktop must run for this to work and the local python/napari environment must be set up. Hopefully this is clearer in the updated installation instructions.

For Linux (<https://github.com/schienstockd/cecilia>):

First, the authors failed to claim they directly start in R environments. They have to say it in the documentation otherwise users without coding experience may not be able to run any line. Second, the authors failed to claim what is the requirement of R version. I am using Windows Subsystem for Linux, Ubuntu 22, to test this, but when I downloaded the default version of R and up-to-date version of R studio server edition and run this line “remotes::install_github("schienstockd/cecilia", Ncpus = 4, repos = "<https://cloud.r-project.org>")” to install, it turns out error because a few packages are not able to install due to conflicted versions.

R-package installations can vary vastly between systems. We therefore opted to utilise ‘renv’ to create a more reproducible environment. The user can download the ‘renv.lock’ file and then call `renv::init()` to restore the package versions. On Linux, this will take some time as the packages will be compiled by source. There are also other system packages that must be installed. These depend on the Linux distribution but R will show in the error message which packages must be installed. We tested this on Ubuntu 24.04.1 LTS and the following packages must be installed with ‘apt install’. As there are a plethora of Linux distributions, we give these general guidelines for Debian based systems and will gladly respond to user requests if needed for other distributions:

- libcurl4-openssl-dev
- libxml2-dev
- cmake
- libgdal-dev
- libmagick++-dev
- libudunits2-dev
- libharfbuzz-dev
- libfribidi-dev

For “Macs” (<https://github.com/schienstockd/cecilia>):

I am actually able to follow their Macs installation approach on my Windows. It is somehow interesting because the authors claimed “This package currently only works on Unix systems because shiny requires this architecture”, and therefore Windows version is only accessible through docker. However, I search the Shiny documentation and asked ChatGPT, surprisingly found that Shiny should be able to run on Windows without any problem. So I directly tried to run this process on my Windows, and did not find any problems related to Shiny. What really stopped me is an error because windows does not support a python library “glmnet” when I was running “`cciaCondaCreate(envType = "image")`”. However, I think this is avoidable, because (1) it is a non-maintained library so the authors may swap it; (2) its function can be altered by other libraries; and (3) R has the same library that is actually supported by windows so you may use it instead.

We removed the ‘glmnet’ dependency. This was pulled from ‘pstree’ which we use to manage module tasks. ‘pstree’ must be available on the host system as a command for module tasks to work. This is standard on Linux systems but not on MacOS. We referred to this is in our documentation and recommend installing it via homebrew.

Therefore, I strongly require authors to check again whether Windows are able to use Shiny and hence Cecelia. If Shiny is indeed available, there should be an easier way for us Windows users to install it, either through Dockers or Bash or R command lines.

Shiny and most other components will work on Windows; however, other packages such as the ‘parallel’ package and its ‘mcparrallel’ function are not available (from the function description: ‘These functions are based on forking and so are not available on Windows.’). We use other command-line tools that are common for Unix systems to work with files. We should have been clearer in our description of that. We envision that we can expand on native Windows support, but we cannot deliver this at this stage. We have added a comment in ‘Running Cecelia with Docker’:

“This setup is currently required for Windows users because packages such as the ‘parallel’ package and its ‘mcparrallel’ function are not available for Windows systems. We further use other command-line tools that are common for Unix systems to work with files which are not native to Windows systems. Future versions will seek to expand on native Windows support”

Therefore, the authors need to carefully inspect and edit their GitHub documentation as well as the app deployment and installation, and finally provide easy and accessible approaches to install Cecelia for potential users with different background and devices. While I certainly believe Cecelia is a strong and impactful toolbox that deserves to be published in Nature Communications, I cannot support the publication with the current documentation until the authors make the installation clear with necessary details, without bugs and misleading info, and accessible to all types of users. While this may be regarded as a major concern, I believe it is not difficult to solve. I would recommend editors to learn from authors if they are able/willing to handle the accessibility issues, to determine the editorial decision.

We thank the reviewer again for the encouraging comments; we have developed substantially improved installation instructions for different operating systems to improve the accessibility of the software, which are available on readthedocs, which we link through the Cecelia Github page: <https://cecelia.readthedocs.io/en/latest/index.html>.

There are some minor issues listed below for authors to consider:

Line 39-42 “current approaches ... each step.”: Lack of detail. You would like to add information describing how complicated and laborious the previous approaches are. For example, what is a typical classic approach pipeline like? How many hours of user’s effort does it take to finish the script and process one sample?

We have added a statement about a typical pipeline for histo-cytometry that provides context for the reader of a current approach:

“For example, a typical histo-cytometry pipeline for 3D images⁹ involves segmentation in either Imaris (Oxford Instruments) alone (0.5-4 h), or in combination with ImageJ for densely packed cells (2-24 h). These data must be exported as CSV files and imported into a flow cytometry package such as FlowJo (BD) for gating (2-4 h). For spatial quantification, the gated populations can be imported into CytoMAP to apply appropriate analysis methods⁸. Once the data is loaded into FlowJo, there is no connection back to the original image and the user cannot visualise spatial analysis directly on the image. Some of these shortcomings have been addressed by the

commercial software suites as StarataQuest (TissueGnostics), however this only support 2D images.”

Line 78-80 “Furthermore ... Python notebooks.” This is extremely good and necessary, but you need to provide detailed explanation in your website documentation about how to enable this. For example, how to call data from your pipeline? How to make customized module and implant into Cecelia processing pipeline?

We recommend starting custom analysis in an R-Markdown and if this proves to be useful for general usage you could expand this to a generic module. Python is used primarily for image processing and some clustering approaches, such as Leiden clustering. We have added a tutorial on this in our documentation:

A) Analyse data in RStudio:

https://cecelia.readthedocs.io/en/latest/access_data_rmarkdown.html

B) Generate a custom module:

https://cecelia.readthedocs.io/en/latest/create_custom_module.html

There is also an element of linking the data that is generated in our framework to other packages. We have tried to demonstrate this with SPIAT¹, spatstat (<https://spatstat.org/>) and celltrackR²: <https://cecelia.readthedocs.io/en/latest/others.html>

Line 99 “Cellpose ... (cyto2)”. Please provide a reference for this model.

We have included this reference. Cyto2 is one of the standard Cellpose models.

Line 98-102 “Segmentation ... analysis.”. Unclear/confounding concept? What do you mean by “segmentation” here? In my dictionary, “segment” in the field of image processing means one connected area of pixel/voxel labeled with certain identity in the image. In this case, it means each identified cell. Accordingly, “segmentation” means process to identify and label individual cells in each channel.

Yes, that is true.

If this is true, then what do you mean by “segmented channels were then merged into a combined segmentation”? I think “segmented” channels means some type of cell labeling; but if you combine them together, should they be called “segmentation” anymore? It would be just a total labeling dataset. Or do you mean something totally different from what I am thinking about? Please specify.

We segment the channels individually then merge the segmentation results so that all segmented objects are combined into one channel. Yes, this is the total labelling dataset, but this term may be confusing. We thought it simpler to refer to this as a ‘combined segmentation’. In the app the user can define this order. For example, we can segment dendritic cells first and then T cells. In this way the user can keep the segmentations separate and find the optimal settings for each channel. Here is a screenshot how this looks in the app:

Line 112. What is the full name of IBEX? Also please provide a very brief introduction, perhaps plus ref.

IBEX refers to ‘iterative bleaching extends multiplexity’. We have added the following paragraph:

“IBEX is a method to increase the number of fluorophores that can be imaged on samples by performing iterative staining and bleaching³. This requires a common marker across imaging cycles for subsequent image registrations, for example a nuclear marker such as Hoechst or DAPI. Current protocols use custom scripts and the commercial Imaris software for analysis and SimpleITK (simpleitk.org) for channel alignment³.”

Line 117. Please provide a brief introduction of Leiden clustering because not all readers understand this concept. Perhaps also provide reference. Correspondingly, in Fig. 3A, please describe in the legend what x-axis means in your Leiden clustering.

We have added the following paragraph:

“Leiden clustering is a community detection algorithm⁴ that attempts to maximise the difference between groups while maintaining local cohesion between elements.”

We also added the following explanation in the figure legend of Fig. 3A:

“Identified clusters are represented along x-axis.”

Line 125. Same issue. What is the full name of CODEX? Also please provide a very brief introduction, perhaps plus ref.

We have added the following paragraph:

“To further explore functional regions, we examined CODEX (co-detection by indexing)^{5,6} images from healthy human patients (portal.hubmapconsortium.org) (Figure 3b). CODEX utilises DNA-conjugated antibodies and iterative imaging with chemical stripping to enable imaging of large marker panels⁵.”

Line 130. Ref for Delaunay triangulation?

We have added a reference for this method: ⁷.

Line 125. Same issue. Please provide a very brief introduction plus ref for Xenium in situ analysis.

We have added the following paragraph:

“Xenium is a fluorescent imaging technology that cycles through iterative DNA probe ligation to RNA targets, imaging and probe removal⁸.”

Line 151-155. Please provide a very brief biological background for people without corresponding immunological knowledge.

We have added the following paragraph:

“Upon infection of the skin or mucosal epithelium, HSV undergoes local replication and is controlled by virus-specific CD8⁺ and CD4⁺ T cell responses⁹.”

Line 151. Strange sentence. An easiest way is to change “imaged” to “found”.

We changed this as suggested.

Line 164. Recommend a ref for HMM.

We have added references: ^{10, 11}

Line 166. Will you consider changing the terminology “angle”? This name does not reflect the concept you are measure. If my understanding is correct, you are measuring the cumulative angle of change of direction of the cell, but generally when general reदार hear “angle”, we cannot reflect on cell movement there is no absolute coordinate system and cell always change direction. Therefore I highly recommend you change this terminology to “cumulative change of direction” or whatever similar.

We measure the angle between two lines which are defined by three points. That means that the first 2 objects within a track do not have an angle and the first object within a track does not have speed. We have added a section how to create a custom module where we go through the use-case to define a user defined cumulative angle change of direction:

https://cecilia.readthedocs.io/en/latest/create_custom_module.html

Line 190. What do you mean by “segmentation” here? I don’t understand.

This is about combining the segmentations of differently sized cells and structures into one segmentation that we can use for subsequent gating and clustering. If we keep these segmentations separate, then there could be an issue that one fluorescent signal is segmented twice and then counted twice.

Fig. 2-5. For many microscopic images you labeled in the top left with 3 colored circles, does that mean the total fluorescent dye or tested mRNA number you involve, i.e., channel number? Whatever it means, please make sure you indeed describe it in the legend.

Yes, the circles represent the number of channels. We have added to the figure legends descriptions of what these numbers are.

Also, for those 2D/3D labels, I strongly recommend you change them to x-y, x-y-z, x-y-t, or x-y-z-t, because readers need to easily understand the data structure you are using. When people see “3D”, they don’t understand if you are talking about voxel image or video.

We have added these as suggested.

Line 199-201. Please provide a very brief biological background to let readers understand what you are doing. Please also mention the word “silence”, so that more general readers can easily reflect that shRNA is an approach to silence target gene in your mammalian cell system.

We utilised this published data to show that we can process an array of 2D timelapse images to extract migration properties. We added the term ‘silence’ and included an explanation of the assay:

“These assays are used to investigate migration speed and dissemination properties of cancer cells in different settings.”

Line 202. It is a good idea to apply ILEE anisotropy function in this way, but when you were using ILEE anisotropy function, did you adjust tensor box size? ILEE is designed for cytoskeleton in a single cell, whose tensor box size by voxel may not apply to your usage looking at large amount of cell group. So, you may need to adjust to make your measurement accurate and reflecting your feature appropriately.

Yes, we have a parameter that the user can define to account for the granularity of the signal. For most of the images we used a radius of 50 μm .

Fig. 2B. Please use the same field of view for each image, or mark which part in the merged image you used to show the split channel.

We have amended this to show the same field of view.

Fig. 4A. Image presentation is misleading. I think you need to mark on panel A that this is x-y-t or x-y-z-t video and also state in the legend that this representative frame of your video which is not looking the same area of the same sample at different time point.

We have added the following paragraph:

“Representative frames of gDT-II CD4⁺ and gBT-I CD8⁺ T cells at different timepoints of infection: naïve mice and 1d and 2d after infection”

Reviewer #1 (Remarks on code availability):

I spent 6 hours and failed to installed it. Overall accessibility is their only major issue, so I wrote remarks over the app in the main comment, as above. Please check.

We certainly apologise for this difficulty. We have streamlined the installation, improved the instructions available online, and had many users of different skill levels independently test the installation of Cecelia. Our new instructions will enable users to solve issues, and we are committed to providing support and continuing to improve the software.

Reviewer #2 (Remarks to the Author):

In this manuscript, Schienstock et al present a new open-source image analysis toolkit, Cecelia, that integrates multiple existing pipelines into a unified platform. In addition to advanced analysis of static images, Cecelia provides the ability for interrogation of time-resolved data, such as by incorporating dynamics of cell migration with Hidden Markov Models to allow advanced classification of cell behavior with respect to tissue location. Cecelia is not entirely novel in itself, as it is an integration of previously developed algorithms and pipelines. However, this integration is important as it generates a tool and resource for the bioimaging community which will help with standardized quantification of diverse imaging datasets.

The manuscript is generally well-written. Detailed instructions for installation and use are also provided in the supplementary material and on a separate website, making this platform relatively easy to use/test.

The main concern is regarding the level of detail provided to the reported analyses and results. Most figures and accompanying text are very general and lack sufficient detail to understand the specifics of how the results were generated or what exactly is shown. Here are some specific areas that could use improvement in reporting:

- Figure 2D: How were clustered vs non-clustered T cells defined? What is meant by the term “in contact with”?

We have added the following paragraph to clarify how clustered cells and cells in contact were defined:

“Clustering P14 T cells were identified using DBSCAN¹², a density based clustering algorithm that identifies groups of coherent objects within a certain diameter (20 μm) and minimum number of elements (4 cells). An interaction between two cells was defined as the closest neighbouring cell from the Delaunay triangulation.”

- Figure 3A: How were B cell rich, T cell rich, and Macrophages regions defined (manual vs algorithmic)?

For this analysis, we used K-Means with 3 clusters. We have added the following paragraph: “Cell regions were extracted by firstly generating cellular neighbourhoods using a radius of 50 μm around each cell. These neighbourhoods were clustered into 3 regions using K-Means¹³ clustering resulting in B cell, T cell and macrophage rich regions.”

- Figure 3B: is the UMPA clustering showing cell clustering or region clustering? For the spatial image of tissue regions, is it cells or regions which are shown? If cells, then this is not analysis of tissue regions, rather of cell type distribution. As above, if the data show cell type clustering, then how were the regions identified?

These are regions based on the results of the Leiden cell clustering. We extracted regions to show more clearly the architecture of the lymph nodes and to highlight our ability to extract these across multiple images in batch. We have added the following paragraph:

“To demonstrate the applicability of Cecelia for analysis of multiplexed 2D images, we segmented 14 CODEX images of human lymph nodes and used Leiden clustering with subsequent Delaunay triangulation⁷ to extract cellular neighbourhoods. These neighbourhoods were subjected to K-Means clustering to extract distinct regions enriched in T and B cells as well as lymphatics and blood vessels (Figure 3b).”

- Figure 3C: same as in 3B - what is shown in each panel, cells or regions? How were region interactions quantified?

We have added the following paragraph:

“We identified individual cell types, as above, using Leiden clustering. Cellular regions were extracted using subsequent neighbourhood detection around each cell with a radius of 100 μm and K-Means clustering to extract 7 regions. Interactions between regions were extracted by mapping the closest cells to each other.”

- Figure 4H-K: It would be helpful to describe in greater detail the workflow and results of these panels. As currently written, the reader has to guess what is shown, how things were calculated, and what the graphs demonstrate.

Thank you for this suggestion. We added the following paragraph:

“We first individually segmented gBT-I CD8⁺ T cells, gDT-II CD4⁺ T cells, XCR1⁺ DCs and migrating TRITC⁺ cells using Cellpose. The resulting objects were tracked using btrack. HMM cannot only be utilised for movement pattern but for any measurement over time. In addition to the movement HMM, as utilised above, we used a combination of morphological parameters to extract 4 shape states (Figure 4i). We utilised the transitions between these states in combination with whole track measurements to deduce track clusters of movement and shape (Figure 4i). These track clusters were able to distinguish DC and T cell behaviour (Figure 4j) where DCs primarily have reduced motility (scanning) or a meandering prolate behaviour. T cells on the other hand exhibit, relative to DCs, more meandering oblate and directed behaviour in addition to their specific aggregation behaviour due to the virus infection (Figure 4k).”

- Figure 5A: What was the cutoff used for determining if cells are close to versus away from lymphatics? It would be good to show two separate UMAP plots or color-coded dots for cells close to and away from the lymphatics to demonstrate how these differ. How does this compare to standard velocity measures?

The distance to the lymphatics was part of the Leiden clustering to generate these populations. There is therefore no cutoff in that sense. We replaced the UMAP with the heatmap to better show the behaviour clustering parameters.

In our view, this analysis could not be done with conventional velocity measures because there would only be one cell population with an average velocity. We have added a comparative analysis with a manual threshold and the corresponding velocity of cells (Supplementary Fig. 3a). With our approach we hope to demonstrate, as in Figure 4a-d, that we can parse the cells into multiple behaviour types based on cell intrinsic and extrinsic parameters in addition to velocity.

- Figure 5B: The text states that MYO10 targeted samples had decreased migration velocity. It is not clear where these data are shown and whether these results are significant. How does this compare to standard velocity measures?

We have corrected this. What we wanted to highlight here is that tracks can also be analysed with collective motion. While the difference between the two treatments is also apparent from tracking measurements, we believe that this way of thinking might add further confidence in the observed phenotype. We also added more comparisons between the two treatments for conventional tracking analysis in Supplementary Figure 3b.

Reviewer #3 (Remarks to the Author):

In this manuscript, the authors present Cecelia, an innovative open-source toolbox designed to integrate advanced quantitative analysis of both static and live cell imaging. The toolbox provides a unified framework for multidimensional image analysis, enabling researchers to handle complex imaging datasets and conduct spatial and behavioral cell analysis with well efficiency. By incorporating Hidden Markov Models (HMM) and real-time image processing with a user-friendly interface, Cecelia offers a powerful solution for studying dynamic immune cell behaviors and interactions. This work significantly contributes to the field of biological imaging by addressing the challenges of analyzing highly multiplexed datasets. However, for a more complete manuscript, we present some supplements and corrections.

Minor Comment

1. In the Introduction section (p.2), it would be helpful to add a brief explanation to clarify the statement “Studies have highlighted that the 3D tissue organisation is critical to understand immune responses” for better reader understanding.

We have added the following paragraph:

“Studies have highlighted that the 3D organisation of tissues, including the lymphoid organs, is crucial for the efficient induction of immune responses¹⁴⁻¹⁸. For example, T cells must traverse complex 3D networks of fibroblasts to locate and interact with cognate antigen presenting cells¹⁹.”

2. Similarly, to facilitate the readers' comprehension, it is necessary to provide an explanation and references in the sentence “Current approaches rely on a sequence of software tools ~” to clarify what specific issues arise from this approach.

We have added the following paragraph:

“For example, a typical histo-cytometry pipeline for 3D images⁵ involves segmentation in Imaris (Oxford Instruments) (0.5-4 h), or in combination with ImageJ for densely packed cells (2-24 h). These data must be exported as CSV files and imported into a flow cytometry package such as FlowJo (BD) for gating (2-4 h). For spatial quantification, the gated populations can be imported into CytoMAP to apply appropriate analysis methods⁴. Once the data is loaded into FlowJo, there is no connection back to the original image and the user cannot visualise spatial analysis directly on the image. Some of these shortcomings have been addressed by the commercial software StrataQuest (TissueGnostics), however this only support 2D images.”

3. In the Methods section, a brief introduction and references to the methods used should be added. For example, methods like Leiden clustering and KMeans clustering should be briefly explained and referenced.

We added these directly into the main text:

“Leiden clustering is a community detection algorithm⁴ that attempts to maximise the difference between groups while maintaining local cohesion between elements. Cell regions were extracted by firstly generating cellular neighbourhoods using a radius of 50 μm around each cell. K-Means clustering¹³ was used to cluster these neighbourhoods into 3 regions. K-Means clustering is a method to partition datapoint based on a predefined number of centre points. The aim is to categorise points by minimising the variance within in each partition. This method resulted in B cell, T cell and macrophage rich regions.”

4. In the Discussion section, it would be beneficial to include a description of Cecelia's limitations and potential future developments.

Thank you for this suggestion. We have added a section at the end of the Discussion:

“Limitations and future development

The use of R/shiny, albeit a straightforward choice for our use-cases (outlined in detail in Methods section), is not the most performant framework for web-applications and its combination with a python backend forced us to create a solution that is more complex to install. Given this complexity and the usage of Anndata as a primary storage system, there can also be race conditions if shiny and python want to access the same dataset at the same time. The shiny modules must therefore be thoughtfully designed when data is used in both environments. While it might be possible to implement the whole framework in python and therefore reduce some of these complications, it might not provide the same functionality as we would need to match processing that is currently being done in R with python functions. Future developments could develop a solution that is a standalone executable based on the tools and processes that we are presenting in this article. Given the rise of artificial intelligence in many areas of image processing and analysis, as tools are developed or improved, our framework can be updated or new modules added. Our hope is therefore to provide a framework to create new interfaces and processing modules for this to be feasible.”

5. In the Result section(p.3), The author mentioned that Cecelia, which was developed, is capable of real-time analysis of resulting data on microscopy images. Does the analysis time vary as the number of cells observed increases? It would be good to add the results of the analysis on the execution time of the cell tracking algorithm.

We did not mean to suggest that we can do on-the-fly analysis of the data. Although we played around with Dask and segmentation on-the-fly, we did not get very far with this approach. What we meant to indicate is that the results are interactive; One issue with other approaches is that in many cases, the downstream analysis is completely disconnected from the image. While there are now quite a few napari plugins to visualise processing results, these are mostly targeted to specific use-cases and the underlying data storage is not cohesive. What we aimed to do in our framework is to always allow the user to view the results back on the image. While there is a lot of development for 2D whole tissue slice imaging in this area, there is not much in the 3D space nor the timelapse imaging field. We therefore hope that this framework could be a starting point to create interactive analysis processing and visualisation independent of the nature of the image.

Regarding execution time for tracking cells, this really depends on how many cells there are, what the search radius is and how many tracks will be detected. There is a post-tracking stage in btrack that will test hypothesis and filter out tracks based on certain criteria. We shouldn't claim it is real-time because it is not – it takes a few seconds. The advantage is just that we can do this in batch for many images and then visualise the results in napari.

6. In the Methods section(p.27), Before performing cell and structure segmentation, a cleanup process was applied to the images, and it is mentioned that this resulted in a cleaner output compared to commercial software such as Imaris. However, there are no comparison results

provided to support this claim. Including such a comparison would make Cecelia's advantages more convincingly presented.

'Cleaner output' was probably the wrong way to word this. The issue that we face with channel subtraction for a larger number of images is that intensity differences vary between images. We can adjust for that by a multiplication factor in Imaris, however this varies between images. The issue is therefore that batch processing is not possible in this manner. The other issue is inhomogeneity across the imaging plane due to different optical properties in the tissue or depth of cells. These can be rectified by using denoising algorithms such as Cellpose 3²⁰ which we used to further refine the data in Figure 4h-k. Due to this denoising strategy we could pick up more cells that appear dim in the image. This resulted in better separation of T cells and dendritic cells during behaviour clustering.

We added the following passage to the Methods:

“Channel subtraction often requires a multiplication factor to account for intensity differences between channels. This factor might vary from image to image slightly. Our channel division method did not require individual adjustments across images which enabled batch processing. This cleanup step can be further improved with the recent Cellpose denoising framework²⁰ to account for intensity inhomogeneity within imaging planes.”

We have also added Supplementary Figure 2 to compare the Cecelia and Imaris approach. They provide similar outcomes. The main difference is that Cellpose's denoising method can have a benefit to detect dimmer cells and make segmentation more feasible.

7. Is there a function that allows for manual correction of errors during the cell tracking process? How do you classify cells with errors in the tracking results?

At the time of submission, we did not have a method to manually correct for tracking errors. We rather filtered objects and tracks that did not fit certain criteria. Especially for T cell migration data, a few tracking errors can be acceptable when there are many cells, and it is often challenging to discern certain tracks. Since submission, we implemented a prototype to detect tracking errors with celltrackR² (<https://ingewortel.github.io/celltrackR/vignettes-out/QC.html>) and correct these using, for the moment, simple joining and deleting methods. We integrated a test version of this functionality into the main app and are planning to expand these capabilities to further support tracking correction when needed.

Below is a screenshot of the current implementation. The user can filter tracks based on quality control measures. These filtered tracks can then be selected and either removed or joined. When the user selects tracks they will be highlighted at the same time in napari. In this way we hope to leverage the analysis capabilities from celltrackR and napari's track modules.

References

1. Feng, Y. et al. Spatial analysis with SPIAT and spaSim to characterize and simulate tissue microenvironments. *Nat Commun* **14**, 2697 (2023).
2. Wortel, I.M.N. et al. CelltrackR: An R package for fast and flexible analysis of immune cell migration data. *Immunoinformatics* **1-2**, 100003 (2021).
3. Radtke, A.J. et al. IBEX: an iterative immunolabeling and chemical bleaching method for high-content imaging of diverse tissues. *Nat Protoc* **17**, 378-401 (2022).
4. Traag, V.A., Waltman, L. & van Eck, N.J. From Louvain to Leiden: guaranteeing well-connected communities. *Sci Rep* **9**, 5233 (2019).
5. Black, S. et al. CODEX multiplexed tissue imaging with DNA-conjugated antibodies. *Nat Protoc* **16**, 3802-3835 (2021).
6. Goltsev, Y. et al. Deep Profiling of Mouse Splenic Architecture with CODEX Multiplexed Imaging. *Cell* **174**, 968-981 e915 (2018).
7. Lee, D.T. & Lin, A.K. Generalized Delaunay Triangulation for Planar Graphs. *Discrete Comput Geom* **1**, 201-217 (1986).
8. Janesick, A. et al. High resolution mapping of the tumor microenvironment using integrated single-cell, spatial and in situ analysis. *Nat Commun* **14**, 8353 (2023).
9. van Lint, A. et al. Herpes simplex virus-specific CD8+ T cells can clear established lytic infections from skin and nerves and can partially limit the early spread of virus after cutaneous inoculation. *J Immunol* **172**, 392-397 (2004).
10. McClintock, B.T. & Michelot, T. momentuHMM: R package for generalized hidden Markov models of animal movement. *Methods Ecol Evol* **9**, 1518-1530 (2018).
11. Visser, I. & Speekenbrink, M. depmixS4: An R Package for Hidden Markov Models. *Journal of Statistical Software* **36**, 1 - 21 (2010).
12. Hahsler, M., Piekenbrock, M. & Doran, D. dbscan: Fast Density-Based Clustering with R *Journal of Statistical Software* **91**, 1-30 (2019).
13. Hartigan, J.A. & Wong, M.A. A K-Means Clustering Algorithm. *Journal of the Royal Statistical Society. Series C (Applied Statistics)* **28**, 100-108 (1979).
14. Leal, J.M. et al. Innate cell microenvironments in lymph nodes shape the generation of T cell responses during type I inflammation. *Sci Immunol* **6** (2021).
15. Hor, J.L. et al. Spatiotemporally Distinct Interactions with Dendritic Cell Subsets Facilitates CD4+ and CD8+ T Cell Activation to Localized Viral Infection. *Immunity* **43**, 554-565 (2015).
16. Duckworth, B.C. et al. Effector and stem-like memory cell fates are imprinted in distinct lymph node niches directed by CXCR3 ligands. *Nat Immunol* **22**, 434-448 (2021).
17. Gebhardt, T. et al. Different patterns of peripheral migration by memory CD4+ and CD8+ T cells. *Nature* **477**, 216-219 (2011).
18. Grootveld, A.K. et al. Apoptotic cell fragments locally activate tingible body macrophages in the germinal center. *Cell* **186**, 1144-1161 e1118 (2023).
19. Ugur, M. & Mueller, S.N. T cell and dendritic cell interactions in lymphoid organs: More than just being in the right place at the right time. *Immunol Rev* **289**, 115-128 (2019).

20. Stringer, C. & Pachitariu, M. Cellpose3: one-click image restoration for improved cellular segmentation. *bioRxiv*, 2024.2002.2010.579780 (2024).

NCOMMS-24-58473A

Response to Reviewers.

We thank all three Reviewers for feedback and supportive comments.

Reviewer #1 (Remarks to the Author):

Same as previous opinion, the paper should significantly contribute to related scientific community that demands powerful and accessible image computation toolbox. Below is my updated opinion on the current renewed submission:

1. All minor issues are fixed in the main text and figure, at least to a level that I don't have concern.

2. They indeed rewrote the GitHub documentation. Significant improvement. Now I am finally able to successfully install Cecelia to my Windows platform using Docker. However, I still spend 2 hours to debug installing as there are still a few flaws, or misleading content, while some are not necessarily attributable to the authors:

(1) Miniconda and Anaconda are not identical. However, official page of "install Miniconda" leads users to actually install Anaconda, which generates a huge problem here. Both work, but their default environment PATH folder is different, and the users might misunderstand as they indeed installed Miniconda, using default setting. But the authors say in there documentation:

"If you installed Miniconda in a custom location, ie/ not your user account, you must specify that directory in the .command or .bat file by editing the file in a Text editor." If users don't know what is going on, they will think they are using default Miniconda PATH and do not change anything. But the real PATH for Anaconda, which they are directed to install, is:

C:\Users\xxx(my user name)\anaconda3

Therefore, the install .bat will NEVER work. So you should provide additional instruction that following the "install Miniconda" page leads to Anaconda but not Miniconda, and provide instruction how to change their CONDA_DIR.

We are sorry for the confusion. The 'install Miniconda' link points to <https://docs.anaconda.com/miniconda/install/> which is the official website to install Miniconda where it says: "On anaconda.com/download, register with Anaconda, scroll down and select the 64-bit Windows Miniconda installer." We have updated the Docker documentation (https://cecelia.readthedocs.io/en/latest/docker_installation.html) on point 2:

"Install the latest Miniconda (<https://docs.anaconda.com/miniconda/miniconda-install>) version. Be sure to install Miniconda NOT Anaconda as the installation path will be different for the subsequent steps."

(2) They failed to tell that when you run cecelia-Windows-docker.bat, you must have

Windows Docker app opened. Again, not all ppl have used independent Docker pack outside Docker app; also, not all ppl have used Docker by any means. If I didn't feed their total documentation and my bug report to GPT-o1. I would never believe I waste more than 1 hour on such a ridiculous issue.

We updated our documentation

(https://cecelia.readthedocs.io/en/latest/docker_installation.html) on point 6:

“Start Docker Desktop and retrieve Cecelia container. Open the Docker Desktop application and run (or build) the Docker container with `cecelia-MacOSX-docker.command` (Mac) or `cecelia-Windows-docker.bat` (Windows) located in `D:\Public\Cecelia\GIT\ceceliaDocker\`. This will start the local napari environment, retrieve the Docker container during the first run and start the app.”

3. Since I conduct different type of research, I don't have appropriate sample to test performance. But the Cecelia app build quality looks high.

Although there are still concerns on GitHub Documentation, I agree to publish because Cecelia is indeed accessible now, with problem solved. However, I recommend the authors edit their GitHub to solve the problems sooner than later.

Reviewer #1 (Remarks on code availability):

I didn't review detailed code, but checked their installation documentation carefully. Comment as above.

Reviewer #2 (Remarks to the Author):

The authors have adequately addressed all the raised concerns, and I have no further comments. Congratulations on an excellent paper and platform.

Reviewer #3 (Remarks to the Author):

The authors have addressed the reviewer's comments effectively, and the revisions have substantially improved the overall quality of the paper. I believe the manuscript is now suitable for publication.

Reviewer #3 (Remarks on code availability):

The code is designed to run on a Docker-based environment, and the program is built using an R-based language. It is enough to reproduce the results of the paper.